# Both prey and predator features predict the individual predation risk and survival of schooling prey

Jolle Wolter Jolles[1,2,3]*, Matthew MG Sosna[4], Geoffrey PF Mazué[5], Colin R Twomey[6], Joseph Bak-Coleman[7,8], Daniel I Rubenstein[4], Iain D Couzin[1,9,10]*

[1]Department of Collective Behaviour, Max Planck Institute of Animal Behavior, Konstanz, Germany; [2]Zukunftskolleg, University of Konstanz, Konstanz, Germany; [3]Centre for Ecological Research and Forestry Applications (CREAF), Barcelona, Spain; [4]Department of Ecology and Evolutionary Biology, Princeton University, Princeton, United States; [5]School of Life and Environmental Sciences, University of Sydney, Sydney, Australia; [6]Department of Biology, University of Pennsylvania, Philadelphia, United States; [7]eScience Institute, University of Washington, Seattle, United States; [8]Center for an Informed Public, University of Washington, Seattle, United States; [9]Department of Biology, University of Konstanz, Konstanz, Germany; [10]Centre for the Advanced Study of Collective Behaviour, University of Konstanz, Konstanz, Germany

**\*For correspondence:**
j.w.jolles@gmail.com (JWJ);
icouzin@ab.mpg.de (IDC)

**Competing interest:** The authors declare that no competing interests exist.

**Abstract** Predation is one of the main evolutionary drivers of social grouping. While it is well appreciated that predation risk is likely not shared equally among individuals within groups, its detailed quantification has remained difficult due to the speed of attacks and the highly dynamic nature of collective prey response. Here, using high-resolution tracking of solitary predators (Northern pike) hunting schooling fish (golden shiners), we not only provide insights into predator decision-making, but show which key spatial and kinematic features of predator and prey predict the risk of individuals to be targeted and to survive attacks. We found that pike tended to stealthily approach the largest groups, and were often already inside the school when launching their attack, making prey in this frontal 'strike zone' the most vulnerable to be targeted. From the prey's perspective, those fish in central locations, but relatively far from, and less aligned with, neighbours, were most likely to be targeted. While the majority of attacks were successful (70%), targeted individuals that did manage to avoid being captured exhibited a higher maximum acceleration response just before the attack and were further away from the pike's head. Our results highlight the crucial interplay between predators' attack strategy and response of prey underlying the predation risk within mobile animal groups.

## Editor's evaluation

This study, which uses cutting-edge video-tracking methods to investigate predictors of predators' attack success on schooling fish, will be of interest to behavioural, evolutionary, and movement ecologists. It underscores the potential of experimental high-resolution tracking approaches and the importance of examining the perspectives of both the predator and its prey. The article is a valuable contribution to understanding predator-prey dynamics and their role in shaping the evolution of social grouping behaviour.

## Introduction

A key challenge in the life of most animals is to avoid being eaten. Via effects such as enhanced predator detection (*Lima, 1995*; *Magurran et al., 1985*), predator confusion (*Landeau and Terborgh, 1986*), and risk dilution effects (*Foster and Treherne, 1981*; *Turner and Pitcher, 1986*), individuals living and moving in groups can reduce their risk of predation (*Ioannou et al., 2012*; *Krause and Ruxton, 2002*; *Pitcher, 1993*; *Ward and Webster, 2016*). This helps explain why strong predation pressure is known to drive the formation of larger and more cohesive groups (*Beauchamp, 2004*; *Krause and Ruxton, 2002*; *Seghers, 1974*). However, the costs and benefits of grouping are not shared equally among individuals within groups, and besides differential food intake and costs of locomotion, group members themselves may experience widely varying risks of predation (*Handegard et al., 2012*; *Krause, 1994*; *Krause and Ruxton, 2002*). Where and whom predators attack within groups not only has major implications for the selection of individual phenotypes, and thereby the emergence of collective behaviour and the functioning of animal groups (*Farine et al., 2015*; *Jolles et al., 2020*; *Ward and Webster, 2016*), but also shapes the social behaviour of prey and the properties and structure of prey groups. Hence, a better understanding of the factors that influence predation risk *within* animal groups is of fundamental importance.

Previous work has identified many different, and sometimes contradictory, factors that predict prey vulnerability in groups. Some of the most long-standing theoretical work suggests that, when predators appear at random and attack the nearest prey, predation risk should be highest on the edge ('marginal predation') and front of mobile groups (*Bumann et al., 1997*; *Hamilton, 1971*; *Morrell and Romey, 2008*; *Vine, 1971*). If such predictions play out in the real world, individuals should try and surround themselves with others to reduce their domain of danger, known as the 'selfish herd' effect (*Hamilton, 1971*). There indeed exists empirical evidence of such behaviour in a range of species, with individuals moving closer together at the moment they perceive increased predation risk (*Jakobsen and Johnsen, 1988*; *King et al., 2012*; *Krause, 1993*; *Sosna et al., 2019*; *Viscido and Wethey, 2002*; but see *Sankey et al., 2021*). While there is also evidence from several studies that predation risk is higher towards the edge (e.g. *Krause, 1993*; *Romenskyy et al., 2020*; *Romey et al., 2008*) and front (e.g. *Bumann et al., 1997*; *Ioannou et al., 2019*) of animal groups, the empirical evidence for this is equivocal and seems to be system-dependent. For example, a number of studies report opposite patterns, with predation risk being highest in the group centre (e.g. *Brunton, 1997*; *Hobson, 2011*; *Parrish, 1989*) or towards the back of the group (e.g. *Handegard et al., 2012*; *Krause et al., 2017*). Experiments have shown that individuals in such positions actually have poorer access to salient social information, as well as visual information of what happens outside the group (*Rosenthal et al., 2015*), which could potentially help explain these findings.

Much of the focus in the literature on the spatial effects underlying predation risk has looked at centre-to-edge and front-to-back effects, partly because they are the easiest to measure, and largely concentrated only on a single or a few key potential features (but see e.g. *Lambert et al., 2021*; *Romenskyy et al., 2020*). However, it may be more likely that a spectrum of different factors shapes predation risk within groups. In particular, following Hamilton's selfish herd theory (*Hamilton, 1971*), factors related to the spacing of individuals with respect to nearby neighbours have been shown to be of importance, with studies showing individuals with fewer neighbours or a larger domain of danger experiencing higher predation risk (*De Vos and O'Riain, 2010*; *Lambert et al., 2021*; *Quinn and Cresswell, 2006*; *Romenskyy et al., 2020*). But also individuals' alignment to nearby neighbours (*Ioannou et al., 2012*), and features related to their visual information can be expected to play a role, two factors known to strongly influence how well individuals respond to others (*Davidson et al., 2021*; *Rosenthal et al., 2015*; *Sosna et al., 2019*; *Strandburg-Peshkin et al., 2013*).

The majority of previous research has also focused exclusively on prey behaviour. Like in Hamilton's classic model (*Hamilton, 1971*), the predator is thereby treated as an abstract source of risk and any predator-related features as well the interactions among the predator and grouping prey typically are not explicitly evaluated. This is problematic as the behaviour and attack strategy of predators – such as to ambush, stealthily approach, or hunt groups of prey – may be some of the most influential factors that affect which prey are ultimately attacked (*Hirsch and Morrell, 2011*; *Lima, 2002*; *Stankowich, 2003*), and in turn are a strong selective force in the evolution of prey and predator traits (*Lima, 2002*). Furthermore, while considerable empirical work has investigated in detail how predators attack groups of prey, the differential likelihood to be attacked – rather than actual prey survival – is still often

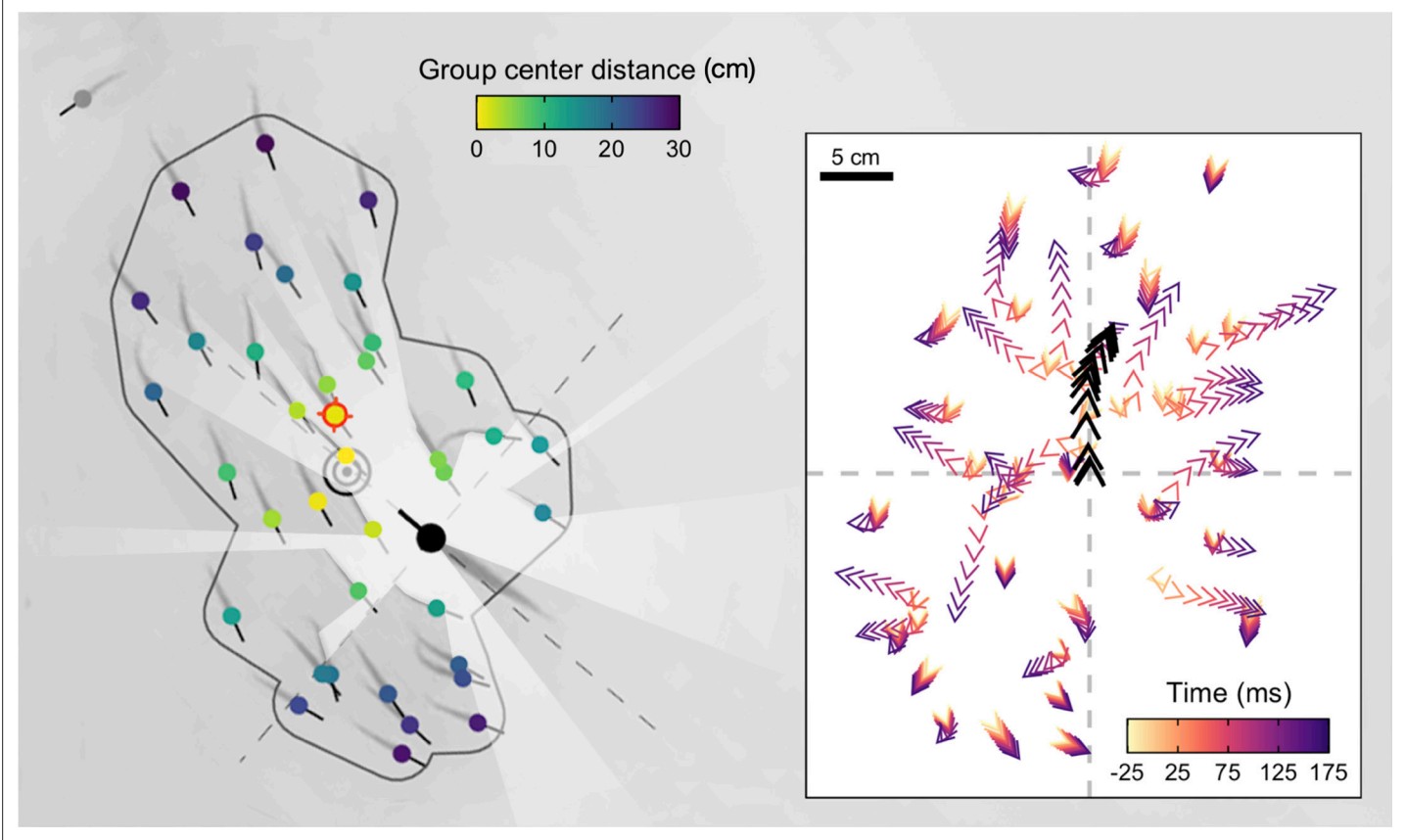

**Figure 1.** High-resolution tracking of predator attacks. Cropped image from a sample video trial moments before the attack with key tracking data overlaid. Shiners are coloured yellow to blue based on their distance from the group centroid. Red target indicates the targeted individual, black concentric circles the group centroid, and the dark grey line the automatically determined school boundary based on hierarchical clustering. Rays (white) represent a visualization of the pike's field of view. Inlay figure presents detailed temporal data of the attack relative to strike initiation, with shiners positioned relative to the pike (black arrows) at the origin facing north.

used to infer predation risk (*Krause, 1994*). This is because of difficulties in quantifying successful attacks in the wild (e.g. *Handegard et al., 2012*) and in the lab (e.g. *Ioannou et al., 2019*; *Milinski, 1977*; *Parrish et al., 1989*; *Romenskyy et al., 2020*). It therefore remains unclear what factors actually affect attack success and the probability for targeted prey to potentially survive attacks.

To advance our understanding of differential predation risk in animal groups, we need to systematically investigate the different absolute and relative spatial and visual features of both prey and predator while considering the real-time dynamics between live predators attacking groups of prey they can actually capture. This, however, poses a considerable challenge since it requires the simultaneous tracking of all members of a group of prey as well as the predator at a sufficiently high spatial and temporal scale. Here, we present experiments in which we achieve this. Specifically, to gain a detailed mechanistic understanding of when and where predators attack groups of prey and what predicts individual predation risk and survival, we observed Northern pike (*Esox lucius*), a geographically widespread and ecologically important predator (*Craig, 2008*), attacking free-swimming schools of 40 golden shiner fish (*Notemigonus crysoleucas*). We exposed pseudo-randomly composed groups (controlling for pike exposure) of juvenile shiners (8.5 cm, 95% CI: 6.4–11.5 cm) to individual pike (n=13; 22.4 cm ±1.1 cm) in a large open arena (1.05 m x 1.98 m; 6 cm depth), and used custom-developed tracking software to acquire detailed spatial and temporal data at 120 fps for a total of 125 attacks (see Materials and methods). By tracking both the predator and all prey individually over time, we were able to quantify each fish's spatial position, relative spacing, orientation, and visual field, and analysed their movement kinematics in detail throughout each attack (*Figure 1*). We then quantified in detail how, when, and where the pike attacked. Subsequently, we used model fitting procedures to infer, both from the prey's and predator's perspective, the relative importance of a suite of potential

**Table 1.** Description of features used in our multi-model inference approach for predicting which individuals were targeted and survived attacks.

For a visualisation of the features, see *Figure 3*.

| Feature | Acronym | Description |
| --- | --- | --- |
| Body length | BL | Shiner's body length (cm) |
| Centre distance | CD | Shiner's distance from the group centroid (cm) |
| Centre-edge position | CDrank | Shiner's CD ranked and scaled from 0 (most central) to 1 (least central) |
| Convex hull position | Hpos | Whether a shiner was part of the group hull or not |
| Inter-individual distance | IID | Shiner's median distance to all of its group mates |
| Local misalignment | LMis | Difference in orientation angle (in degrees) between the shiner and its group mates within 10 cm |
| Voronoi area | VA | Area ($cm^2$) around a shiner closest to that individual and not another individual, limited to the boundaries of the testing arena (log-transformed) |
| Limited domain of danger | LDOD | VA limited to a max radius of 10 cm from the shiner (log-transformed) |
| Front-back centre distance | FBCD | Shiners' distance from the group centroid in the plane of the group average orientation (positive values indicate in front of the centroid) |
| Front-back position | FBrank | Shiners' FBCD ranked and scaled from 1 (front) to 0 (back) |
| Visual weighted degree | WDeg | The proportion of each shiner's vision occupied by conspecifics |
| Distance to the pike | PD | Shiner's distance to the head centroid of the pike (cm) |
| Angle to the pike | PA | Shiner's position relative to the pike facing north (degrees), 0° being straight in front and 180° directly behind |
| Orientation to the pike | PO | The relative orientation (head to tail angle) of the shiner to that of the pike |
| Pike vision of shiner | PVS | Pike's field of view occupied by the individual shiner (deg) |
| Target max speed | TMS | Targeted shiner's maximum speed (cm/s) (smoothed data) |
| Target max acceleration | TMA | Targeted shiner's maximum acceleration ($m/s^2$) (smoothed data) |
| Target max turn | TMT | Targeted shiner's maximum orientation change (deg) in the 0.5 s until the time of attack |
| Pike max acceleration | PMA | Pike's maximum acceleration ($m/s^2$) (smoothed data) |

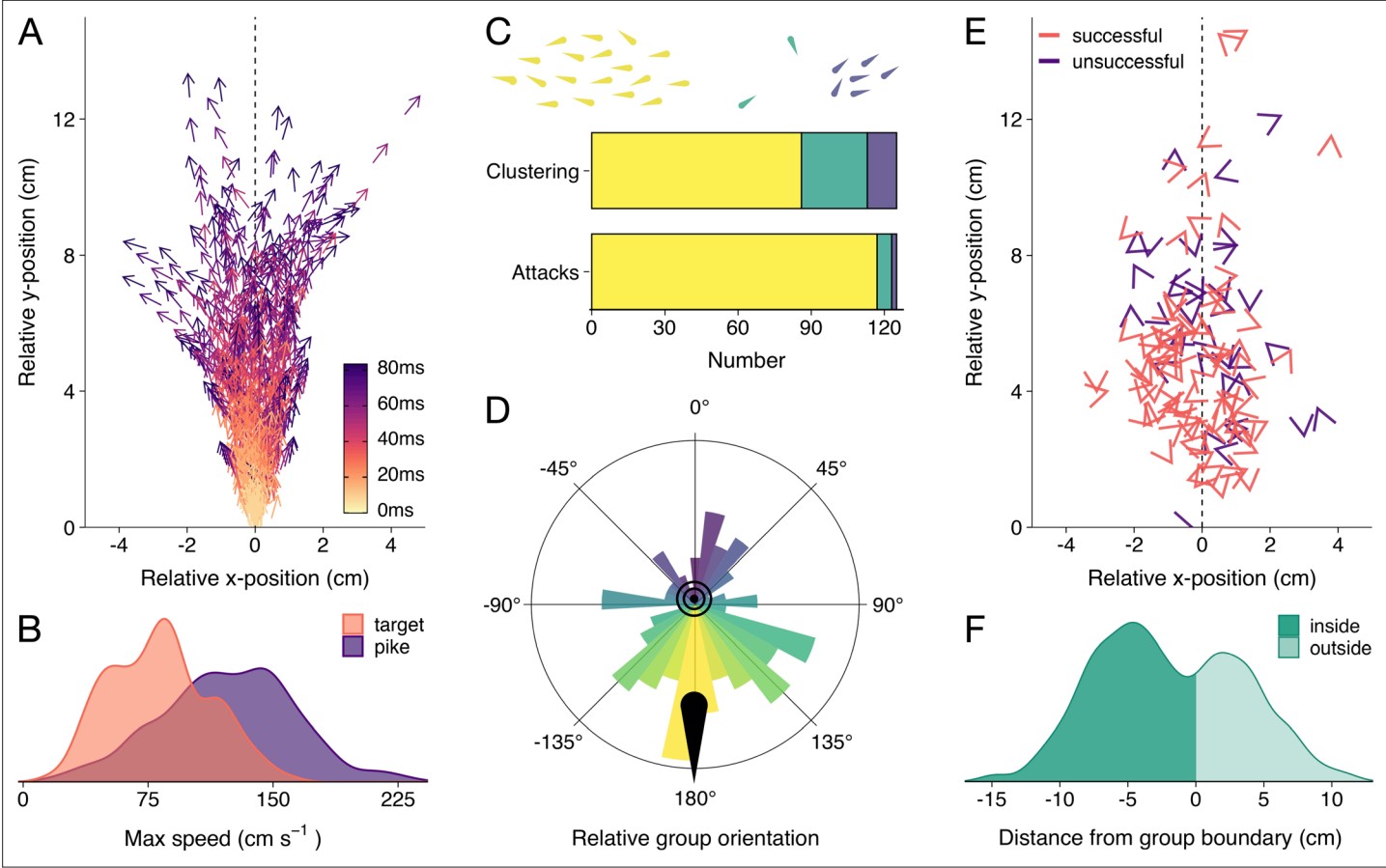

**Figure 2.** Detailed attack characteristics. (**A**) Pike attack trajectories that successfully resulted in prey capture (n=88). Data are shown from the time of attack (strike initiation), with the predator positioned at the origin pointing north. (**B**) Density plots of the maximum (smoothed) speed of the pike and targeted shiners during the attack (from –0.5 s to +0.1 s relative to strike initiation). (**C**) Barplots of shiners' clustering (top) and pikes' likelihood to attack different clusters (bottom). Top bar shows if prey were found in a single cluster (yellow), one large cluster with small clusters of one or two individuals (green), or in multiple larger clusters (blue), as indicated by drawing above, while the bottom bar shows the number of attacks for each type of cluster. (**D**) Polar plot showing the distribution of group orientations relative to the pike pointing north, coloured blue (0°) to yellow (-/+180°). (**E**) Positioning of targeted prey relative to the pike, with arrow headings indicating prey orientations. (**F**) Histogram of pikes' distance from the group boundary. For figure D-F, data were subsetted to attacks of the main cluster (n=117) and focus on the time of attack.

features to predict shiners' risk to be targeted (see *Table 1*). Finally, as pike were able to catch and consume their prey, we were able to investigate what factors best predicted the likelihood of attacks to be successful and thus for prey to survive a predator attack or to be eaten*Table 1*.

## Results

### How do pike attack groups of prey?

The pikes' predatory movements typically began with an orientation phase in which they slowly turned their long body axis towards the schooling prey, followed by a stealthy approach (69% of attacks the pike moved steadily at <0.5 BL/s) during which on average only 5% of shiners turned away (>90°). After getting into position, the pike adopted an S-shaped body posture (see *Appendix 1—figure 1*) to get ready for the actual attack – the strike – one sharp, sudden burst of movement (*Figure 2A*). By curving their body, the pike were able to generate a very rapid increase in kinetic energy (see further *Domenici and Blake, 1997*; *Webb and Skadsen, 1980*) and within a couple milliseconds attain a forward acceleration of 26.7±0.7 m/s$^2$ (mean ± SE), reaching speeds of 122.7±3.6 cm/s, almost 1.5 x higher than the escape speed of the prey they targeted (84.1±2.7 cm/s; $\chi^2$ = 104.7, p<0.001; *Figure 2B*; reported values based on smoothed data). Due to its abrupt nature, we could automatically determine the exact moment of strike initiation at <0.01 s resolution (see Appendix 1), defined as the 'time of

attack', and thus could investigate the individual and collective behaviour of the prey and predator in relation to this exact moment of the attack.

## What is the collective state of the prey at the time of attack?

First, we determined if the shiners were generally found in one large school or multiple smaller groups using a hierarchical clustering approach. In short, fish were automatically clustered based on their inter-individual distance. If a large discontinuity in cluster distances was found, we considered there to be multiple groups of prey, based on a predetermined threshold (see Materials and methods). We found that, by and large, the shiners were organised in one large, cohesive school at the time of attack and rarely showed fission-fusion behaviour (merging and splitting of schools) during the trials. Only occasionally there were one or two singletons besides the main school (25 attacks) or multiple clusters of more than two fish (12 attacks *Figure 2C*), which tended to exist relatively briefly (mean school size: 36.5±0.8). In more than 80% of these cases, pike still targeted an individual in the main cluster (*Figure 2C*). We therefore focused all subsequent analyses on the attacks where the pike targeted an individual in the main cluster, which comprised 94% of all attacks (n=117).

In terms of the collective behaviour of the prey at the time of attack, the shiners were generally highly cohesive (mean inter-individual distance: 16.3±0.4 cm), moderately well aligned (median polarization: 0.59), and moved at a modest speed (mean: 7.7±0.6 cm/s; 90% quantile: 16.3 cm/s, based on the group centroid). Proper rotational milling was rarely observed (mean group rotation order: 0.26), a state that is more characteristic of larger shoal sizes, as observed in similarly-sized experimental arenas (*Davidson et al., 2021*; *Tunstrøm et al., 2013*).

## Where do pike attack schooling prey?

To quantify if pike had a tendency to approach and strike the schools from a certain direction, we computed the groups' centroid position, orientation, and heading (i.e. movement angle) based on all the shiners in the group, and transposed these to be relative to the pike facing north (0°). This revealed that pike had a strong tendency to attack individuals by approaching the groups head-on, both in terms of the groups' relative orientation (circular mean: 170.7°, Rayleigh's test: mean vector = 0.28, p<0.001; *Figure 2D*) and direction of motion (when moving at >1.5 cm/s, n=102; circular mean: 150.6°, mean vector = 0.25, p=0.002). On average, pike launched their attack at 11.5±0.6 cm from the group centroid and only 5.3±0.2 cm from the prey they targeted (*Figure 2E*; 5.46±0.3 cm including attacks where the pike did not attack the main cluster). While the shiners did not show a change in their packing fraction (median nearest-neighbour distance) with repeated exposure to the pike ($F_{1,52}$ = 1.81, p=0.185), they increasingly avoided the area directly in front of the pike's head (*Appendix 2—figure 1*) resulting in the pike attacking from increasingly further away (target distance: $F_{1,52}$ = 45.52, p<0.001, see *Appendix 2—figure 1B and C*).

Ranking individuals front to back (and scaling 1–0), we found that, rather than attacking individuals in the front of the group, pike tended to target individuals in relatively central positions (mean position: 0.45±0.02; n=117). Excluding groups with low polarization (<0.4; n=80), where it is more difficult to determine the 'front', did not substantially change this effect (0.48±0.03). To investigate further if pike potentially launched some of their attacks from inside the school, we computed the smallest convex polygon that encompassed all individuals in the group and used concave approximations to create a realistic approximation of the group boundaries (see *Figure 1*). We found that indeed for more than half the attacks (63.2%), the pike was already partly inside the group boundary at the moment of the attack (based on the location of the head centroid; *Figure 2F*). To investigate if this was by the pike actively entering the group or the group moving to and around the pike we computed the groups' relative motion to that of the pike (see Materials and methods). This revealed that for almost all attacks (92.3%) it was the pike that was responsible for most of this relative movement, with the pike moving more towards the school than the school moving towards the pike.

## Which factors best predict individuals' risk to be targeted in schooling prey?

To infer which features were the most predictive of being targeted for the individuals within the school, we used a multi-model inference approach (for details, see Appendix 3). This is a commonly used technique whereby, rather than fitting a single model, models are fitted for every possible combination

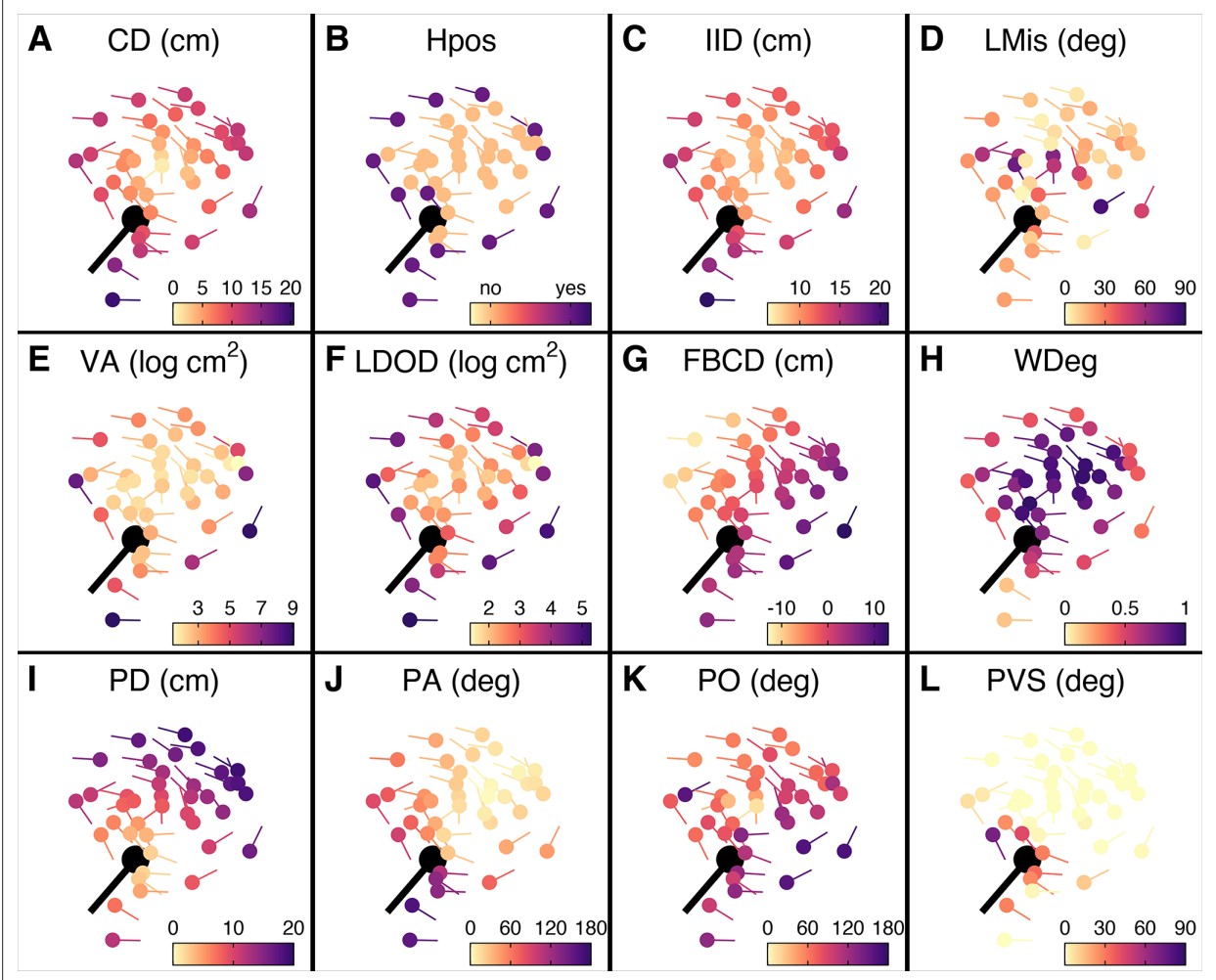

**Figure 3.** Prey features considered in the multi-model inference approach. (A–L) Schooling shiners and pike (black) at the time of attack, with the shiners coloured based on the range of features used in our multi-model inference approach. Visualisations show data for a random representative trial and frame. An explanation of the acronyms can be found in *Table 1*. Note that ranked centre distance (CDrank) and front-back positioning (FBrank) are not included here but will visually resemble plots A and G. Also, for this particular example group rotation was high (0.81) and thus front-back positioning is not meaningful.

of features and their support is ranked based on information criteria (*Grueber et al., 2011*; *Harrison et al., 2018*). Features' importance can then be assessed based on their relative weight across all models, with top-performing models being given more weight (*Burnham and Anderson, 2002*; *Johnson and Omland, 2004*). This approach is not meant to provide evidence for causal relationships but helps to better understand and predict response variables from predictive features. In our models, we considered a combination of features where we had strong biological reasoning to be of potential influence (c.f. *Burnham and Anderson, 2002*), including those related to the spatial positioning, orientation, spacing, and visual field of both predator and prey (see *Table 1* and *Figure 3*). Where relevant, ranked predictors, which place more attention on the relative differences between individuals, were also considered. Input variables were checked for collinearity and otherwise excluded in a step-wise manner (see Material and methods).

## Prey-focused approach

We started with a 'prey-focused approach' whereby we ran multi-model inference using logistic regression considering all individuals in the school as potential prey and excluded any features regarding the predator, thus considering the predator as an abstract source of risk, in line with much previous work (for further details about the models, see Appendix 3). Of the 11 features considered, three

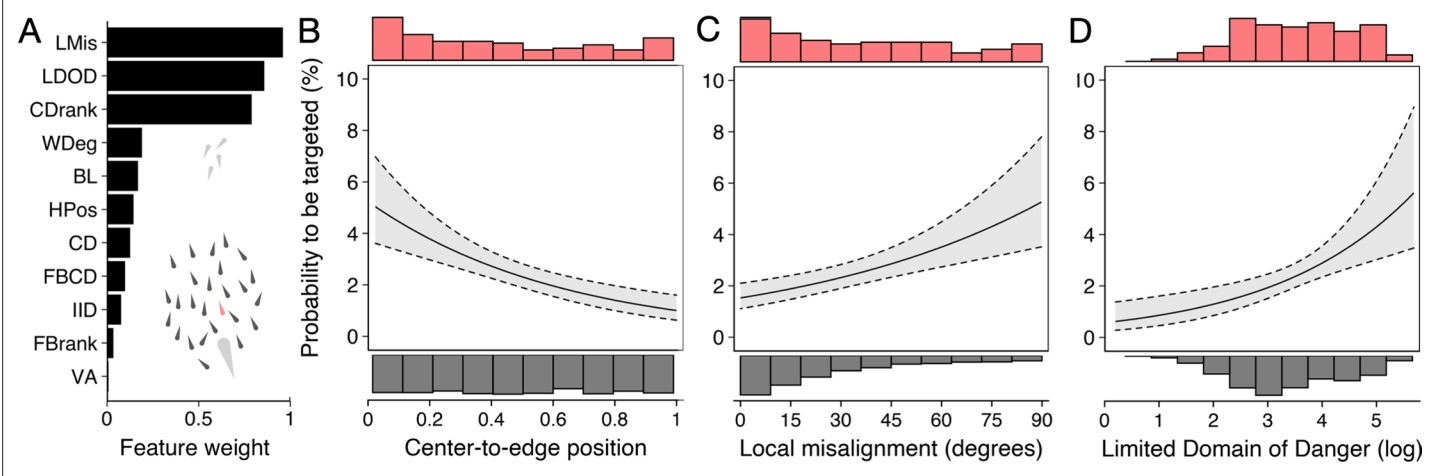

**Figure 4.** Predictors of the likelihood to be targeted - prey-focused approach. (**A**) Relative feature weights based on multi-model inference ranked from highest (top) to lowest (bottom), revealing three key predictive features emerged (for acronyms, see *Table 1*). (**B–D**) Top three features affecting the probability that an individual is targeted: (**B**) its ranked centre-to-edge position, (**C**) its misalignment with nearby neighbours (within 10 cm), and (**D**) its limited domain of danger (log-transformed). Plots are created using predicted values from the final model (see *Appendix 3—figure 1*), with the envelope showing the 95% confidence intervals. Red and grey histograms are of the raw data of the targeted and non-targeted individuals respectively.

key predictive features emerged based on their overall weights (*Figure 4A*) as well as being in the top model (*Appendix 3—figure 1A*) (i) shiner's ranked centre-to-edge position, (ii) shiner's misalignment to surrounding neighbours (within a 10 cm radius), and (iii) the size of shiner's limited domain of danger, the area around a shiner closest to that individual and not another individual, limited to a radius containing on average 25% of the other group members. In contrast to the widely held assumption that predation risk is lowest in the group centre, we found that prey near the centre were more than twice as likely to be targeted than those near the edge of the group (scaled rank 0–1; estimate: –1.69±0.37; LRT: $\chi^2$ = 20.38, p<0.001; *Figure 4B*). Multi-model inference also revealed that individuals were less likely to be attacked when they showed better alignment with their neighbours (estimate: 0.014±0.004 misalignment in degrees; LRT: $\chi^2$ = 15.01, p<0.001; *Figure 4D*) and had a smaller limited domain of danger (LDOD), i.e. were surrounded closely by other groupmates (log area estimate: 0.41±0.12; LRT: $\chi^2$ = 12.99, p<0.001; *Figure 4C*). LDOD was inherently smaller for prey the closer they were to the group centre (correlation coefficient r=0.52), with considerable unexplained variance between these two features (R$^2$=0.28). Prey's front-back position, weighted degree (proportion of vision occupied by conspecifics), or whether they were positioned on the group edge or not were much less predictive of individual's risk to be targeted (i.e. the features were lower ranked, *Figure 4A*). Excluding groups with low polarization did not change the effect of front-back positioning (see Appendix 3).

## Predator-focused approach

By studying predation risk from the prey's perspective, one ignores potentially crucial information about the predator's attack strategy and decision-making (*Hirsch and Morrell, 2011*; *Lima, 2002*; *Stankowich, 2003*). To account for this, we reran multi-model inference but now also considered features about the predator, including shiners' distance, angle, and relative orientation to the pike, as well as the proportion of pike's vision occupied by each shiner (see e.g. *Figure 3L*). As most predators only have a specific region that they are biologically capable of attacking, we also only considered shiners found inside the pikes' strike zone, an area of roughly 8 cm wide and 15 cm long directly in front of the pike within which all targeted prey were positioned (*Figure 2E*). Using this predator-focused approach, we found as most predictive features (*Figure 5A*): (i) prey's distance to the (head of) the pike (–0.514±0.054; LRT: $\chi^2$ = 146.82, p<0.001; *Figure 5B*), (ii) prey's angle to the pike (–0.070±0.011; LRT: $\chi^2$ = 61.32, p<0.001; *Figure 5C*), and, as for the prey-focused approach, (iii) their limited domain of danger (0.675±0.137; LRT: $\chi^2$ = 26.02, p<0.001; *Figure 5D*). Shiners were 6 times as likely to be targeted when they were positioned within 6 cm and directly in front of the pike (-/+45°) compared to when further away or more towards the side (49.1% vs 8.2% of attacks). To investigate if pike actually

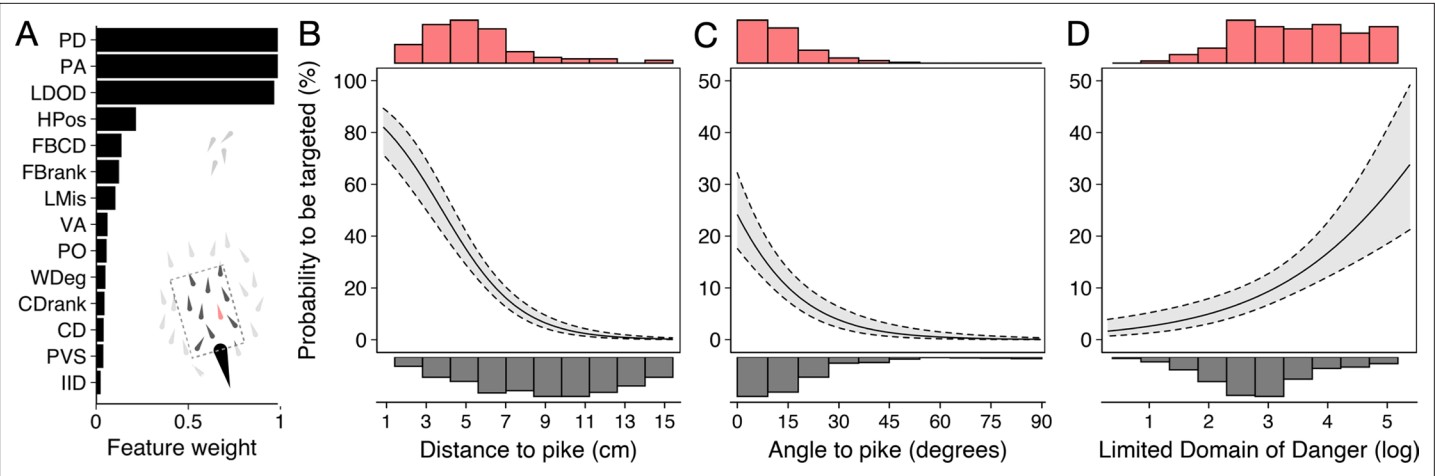

**Figure 5.** Predictors of likelihood to be targeted - predator-focused approach. (**A**) Relative feature weights based on multi-model inference and ranked from highest (top) to lowest (bottom), revealing three key predictive features emerged. (**B–D**) Top three features affecting the probability that an individual is targeted using a predator-focused approach: (**B**) its distance to the pike, (**C**) its angle relative to the pike's orientation, and (**D**) its limited domain of danger. Plots are created using predicted values from the final model (see *Appendix 3—figure 1*), with the envelope showing 95% confidence intervals. Red and grey histograms are of the raw data of the targeted and non-targeted individuals, respectively.

tended to attack the closest prey, one of the assumptions of Hamilton's model (*Hamilton, 1971*), as opposed to generally attacking prey that are near, we ranked individuals based on their distance from the pike, again considering only individuals within the domain of danger. This revealed that for 73% of attacks (86/117) the pike did indeed attack the nearest individual (ahead) and for 90% of attacks one of the three nearest individuals. Consequently, swapping the feature of absolute pike distance with ranked pike distance considerably increased the predictive power of the model (ΔBIC = –37.4), thereby resulting in LDOD to drop as a significant feature.

## Which factors best predict the likelihood for targeted individuals to survive attacks?

The pike in our study were allowed to catch and consume their prey and were relatively successful, with 70% of attacks resulting in prey being eaten. In contrast to previous work that only investigated the likelihood for an individual to be targeted (e.g. *Handegard et al., 2012*; *Ioannou et al., 2012*; *Romenskyy et al., 2020*), we could therefore also assess in detail what features are associated with targeted fish to survive the attack. We compared individuals that were targeted yet successfully evaded capture (n=34) with those individuals that were caught (n=83), and considered as potential relevant features those that were found to be important in predicting which individual was targeted (see above), pike's vision of its target at the time of attack, the targeted individual's maximum speed, acceleration, and turning rate (max change in orientation), and pike's maximum acceleration, all in the same standard 0.5 s time window until the time of attack.

Running multi-model inference as before, two main predictive features emerged (*Figure 6A*; *Appendix 3—figure 2*): (i) shiner's maximum acceleration until the strike (*Figure 6B*) and (ii) shiner's distance to the pike's head (*Figure 6C*), which were themselves only very weakly related (4.9% of variance explained between them). In other words, targeted prey were more likely to evade capture when they showed a quick acceleration response in the moments before the pike launched its attack (–0.12±0.04; LRT: $\chi^2$ = 13.19, p<0.001) and were positioned further from its head (–0.27±0.09; LRT: $\chi^2$ = 8.98, p=0.003).

## Discussion

Understanding when, where, and how predators attack animal groups, and the types of anti-predator benefits grouping animals may experience, has been of long-standing interest (reviewed in *Krause and Ruxton, 2002*; *Pitcher, 1993*; *Ward and Webster, 2016*). Although it is well appreciated that there is differential predation risk within animal groups, our knowledge has, nonetheless, remained

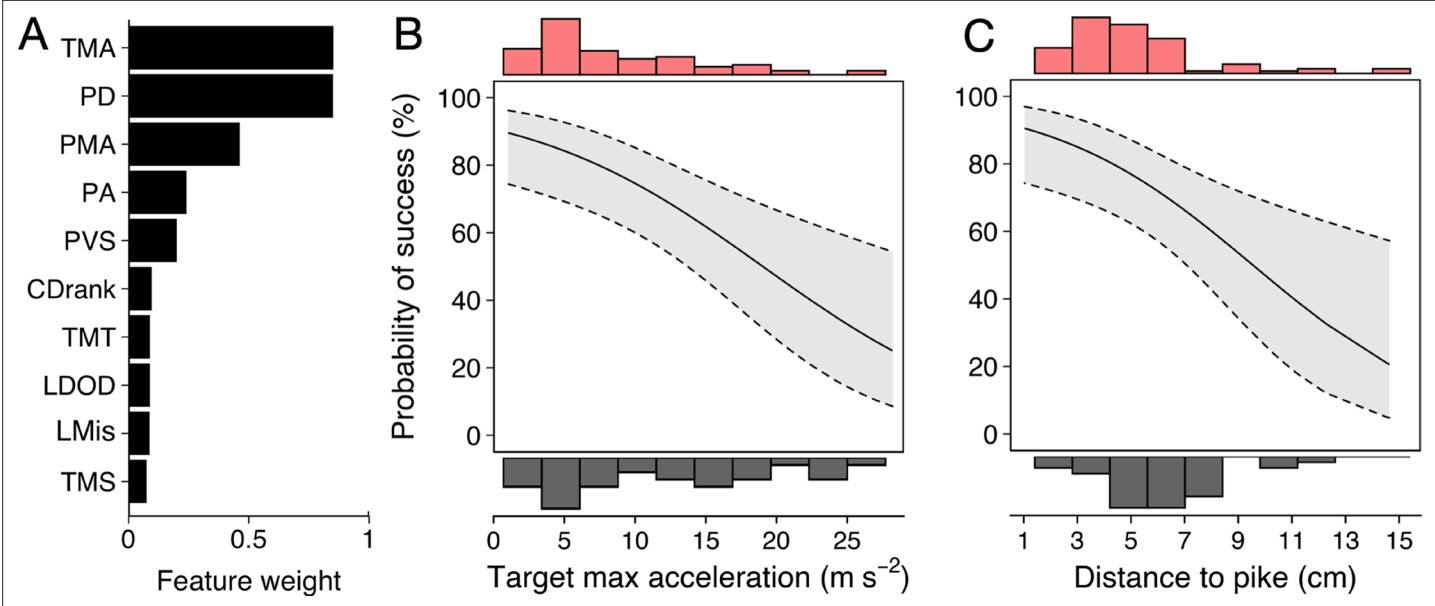

**Figure 6.** Predictors of Predator Attack Success. (**A**) Relative feature weights based on multi-model inference and ranked from highest (top) to lowest (bottom). (**B–C**) The two key features that best predicted predator attack success: targeted shiners' maximum acceleration in the half second before the attack (**B**), and their distance to the pike at the time of attack (**C**). Plots are created using predicted values from the final model, with the envelope showing 95% confidence intervals. Red and grey histograms are of raw data corresponding respectively to individuals that were targeted successfully and those that evaded the attack.

largely centred on marginal predation and selfish herd effects. By employing high-resolution tracking of predators attacking large, dynamically moving groups of prey, here we provide detailed insights into where and when predators attack and reveal a suite of both prey and predator features that best predict the risk of schooling prey to be targeted and survive attacks. Our study shows that consideration of the multi-faceted factors underlying predation risk in combination with predators' attack strategy and decision-making is important to better understand the broader costs and benefits of grouping, with implications for the evolution of social and collective behaviour.

Following a slow and steady approach of the school, pike tended to attack the groups head-on, in one fast burst of movement. However, rather than attacking individuals in the leading edge of the group, often assumed to be more risky than further back in mobile groups (e.g. *Bumann et al., 1997*; *Ioannou et al., 2019*; *Krause et al., 2017*; but see e.g. *Handegard et al., 2012*; *Lambert et al., 2021*), we found that pike often slowly entered the school and then launched their rapid attack towards the group centre. That predators such as pike can get very close to their prey has been described previously (*Coble, 1973*; *Morris et al., 1956*; *Krause et al., 1998*; *Nursall, 1973*; *Webb and Skadsen, 1980*) and could potentially be explained by their narrow frontal profile (*Webb, 1982*) as it makes it very hard for prey to detect movement changes, especially when attacked head-on. This may also explain the suggestion of previous work that pike do not suffer much from the confusion effect (*Turesson and Bronmark, 2004*). That predators may actually enter groups and strike at central individuals is not often considered (*Hirsch and Morrell, 2011*), possibly because it contrasts with the long-standing idea that predation risk is higher on the edge of animal groups (*Duffield and Ioannou, 2017*; *Krause, 1994*; *Krause and Ruxton, 2002*; *Stankowich, 2003*). However, our finding is in line with the predictions of theoretical work that suggest that the extent of marginal predation may depend on attack strategy and declines with the distance from which the predator attacks (*Hirsch and Morrell, 2011*). Furthermore, increased risk of individuals near the centre of groups may be more widespread than currently thought. Predators not only exhibit stealthy behavioural tactics that enable them to approach and attack central individuals, as we show here, but may also do so by attacking groups from above (*Brunton, 1997*) or below (*Clua and Grosvalet, 2001*; *Hobson, 2011*; but see *Romey et al., 2008*), and by rushing into the main body of the group (*Handegard et al., 2012*; *Hobson, 2011*; *Parrish, 1989*).

One of the assumptions of Hamilton's classical model is that the predator appears at random and strikes the nearest prey (*Hamilton, 1971*). This helps explain why, as shown for other systems, that individuals on the edge and front of groups experience higher predation risk (*Bumann et al., 1997*; *Ioannou et al., 2019*; *Krause, 1993*; *Romenskyy et al., 2020*; *Romey et al., 2008*). But although here we also show that pike tended to attack the nearest prey (they were physically capable of attacking), this was importantly not the first prey they encountered. Namely, pike tended to delay their attack until they were very close to and often even surrounded by their prey. Besides potentially having more prey to choose from, pike clearly benefited from this behaviour by being able to generate acceleration and speeding forces considerably higher than that of the prey they targeted (see also *Domenici and Blake, 1997*). This is supported by the finding that attacks were less likely to be successful when the target was further away and thereby able to achieve a high acceleration response in the moments before the strike. That pike very slowly and carefully approached the grouping prey may thus be to minimize their strike distance, position and angle, until the point where they have to decide to either launch the attack or call off it off based on their perceived likelihood for success (*Nilsson and Eklöv, 2008*). This type of predator decision-making *when* to attack, particularly relevant in the context of ambush predators, is rarely considered (*Hirsch and Morrell, 2011*), but fundamental for a proper understanding of the distribution of predation risk in animal groups across different predator-prey systems.

While many studies have investigated differential predation in animal groups, few have systematically compared and considered the many types of features that can be hypothesized to influence predation risk (but see e.g. *Lambert et al., 2021*; *Romenskyy et al., 2020*). Here, using a multi-model inference approach, we found that, in addition to centre-to-edge position, also limited domain of danger and local misalignment were key predictive features of predation risk from the prey's perspective. In line with predictions from Hamilton's selfish herd model (*Hamilton, 1971*), we found that fish with a larger (limited) domain of danger were at higher risk of being targeted. Although some previous studies have shown individual predation risk to be related to their domain of danger (*De Vos and O'Riain, 2010*; *Lambert et al., 2021*; *Romenskyy et al., 2020*), our results highlight the importance of variation in individuals' domain of danger *within* groups. This is further reflected by the finding of considerable unexplained variance between fish's centre-to-edge position and their limited domain of danger. In other words, despite central individuals being more at risk, it still pays for prey to be close to and surrounded by others within the group to reduce their risk of predation. Such spatial heterogeneity within groups and its effects on differential predation is rarely considered (*Jolles et al., 2020*). Our finding that the local alignment of prey was also a strong predictor of predation risk, with less aligned individuals having a higher chance of being targeted, is in line with experimental work using virtual prey (*Ioannou et al., 2012*). Less aligned individuals may be predated more simply by those individuals standing out from their surrounding group mates (i.e. the oddity effect; *Landeau and Terborgh, 1986*) and/or because predators may have evolved to target misaligned individuals because of being less capable of obtaining salient social information (*Couzin and Krause, 2003*). Together, these findings bring some important nuance to the 'selfish herd' phenomenon: rather than moving towards the group centre, individuals should try and position themselves near others, also inside the group, and even when it is of small or medium size, and make sure they do not attract attention based on their orientation.

By rerunning multi-model inference with a predator-focused approach, that is by also including features about the predator and only considering prey within its perceived attack region, we found prey's distance and angle to the pike were the strongest determinants for being targeted (see also *Romenskyy et al., 2020*). This shows that, besides moving towards others, as discussed above, it pays for prey to move away and avoid the cone of risk directly in front of the predator. By repeatedly testing the shiners with the pike, we saw that shiners indeed increasingly distanced themselves from the predator and especially avoided the region in front of the pike. This is consistent with previous work that suggests that, in addition to unlearned predispositions, experience with predators is important (*Kelley and Magurran, 2003*). While much previous work has only focused on predation risk from the prey's perspective, these findings highlight that, for a proper understanding of predation risk in animal groups it is important to not remove the predator from the equation (*Hirsch and Morrell, 2011*; *Lima, 2002*).

Vision is known to be the primary modality for pike's predatory behaviour (*Nilsson and Eklöv, 2008*) and for mediating social interactions among species of schooling fish (*Kotrschal et al., 1998*; *Rosenthal et al., 2015*; *Strandburg-Peshkin et al., 2013*). We however found that the extent that a shiner's vision was occupied by conspecifics, the extent it could see the predator, and the extent the predator could see the shiner were only weak predictors of individual predation risk. This does not mean they are not relevant per se, as they were closely linked to the main predictors, but, being more specific, could potentially explain less of the observed variance overall. Hence, future experiments specifically manipulating the conditions that impact the vision of predator and prey would be valuable in further disentangling the role of visual information in predator-prey interactions. Furthermore, while we unravelled key features that are strongly predictive of which individuals were most likely to be targeted for an attack, further future work, including experimental manipulations and/or directed acyclic graphs (DAGs) (*Laubach et al., 2021*; *McElreath, 2020*), is also needed to properly disentangle cause and effect in the predation risk of grouping prey.

Overall, the pike were very successful, with 70% of attacks resulting in the targeted shiner being eaten. This is comparable to previous studies with sit-and-wait predators (*Krause et al., 1998*; *Neill and Cullen, 1974*; *Turesson and Bronmark, 2004*; *Webb and Skadsen, 1980*). But where previous work was not able to quantify differential predation in terms of mortality risk, such as by using confined or virtual prey (*Ioannou et al., 2019*; *Milinski, 1977*), difficulties of field conditions (*Handegard et al., 2012*), or by the predator simply failing to ever attack successfully (*Parrish, 1989*; *Romenskyy et al., 2020*), here we were able to also investigate the factors that may affect the likelihood for an individual to survive an attack. We found that both predator distance and prey acceleration in the moments until the attack were highly predictive of predation success, with targeted prey that were further away, and that showed a faster acceleration response, being more likely to evade capture. These two features have also been found as significant variables affecting survival in a study on predator-prey dynamics with single prey (*Walker et al., 2005*; see also *Lucas et al., 2021*). Together, this suggests that targeted prey could sometimes anticipate the strike, highlighting that, despite the very high speeds pike attained, prey response does matter in predator-prey dynamics, and that evasive prey behaviour may be especially successful when prey are further away (see also *Ranåker et al., 2012*).

Although other ambush predators may employ different attack strategies, and even among pike there may be variation in attack behaviour, the principles we highlight related to the attack sequence and when, where, and how predators attack are relevant for other systems. One thing however that is not clear yet is how the observed effects play out in much larger prey groups. In particular, while group size is not expected to effect much whether ambush predators are likely to attack internal individuals, the specific risk of central individuals could both be hypothesized to decrease with group size, such as if the predator is more likely to attack when surrounded by prey, or to not be affected by it, such as if the predator actively targets central individuals. Whatever the process, the observed findings are likely for prey that move in groups of somewhat intermediate size; for very large groups, such as the huge schools encountered in the pelagic, ambush predators may simply not be able to attack the group centre due to spatial constraints. More generally, the tendency for predators to attack the centre of moving groups may depend on the medium in which the predator-prey interactions occur. As in the air there is potential for (fatal) collisions, and on land it is physically difficult for predators to enter groups and predators' size advantage tends to be more limited, predators may be less likely to go for the group centre as compared to in aquatic or mixed (e.g. aerial predator hunting aquatic prey) systems. Hence, the important interplay we highlight between predator attack strategy and prey response may have different implications across different predator prey systems and warrants concerted further research effort.

Predation is seen as one of the main factors to shape the collective properties of animal groups (*Herbert-Read et al., 2017*) and has been shown to drive the formation of larger, more cohesive groups that exhibit collective, coordinated motion (see e.g. *Beauchamp, 2004*; *Ioannou et al., 2012*; *Seghers, 1974*). Our finding that central individuals are more at risk of being predated could actually have the opposite effect, with schooling having a selective disadvantage, which over time could result in weaker collective behaviour and less cohesive schools. However, we do not deem this likely as selection is likely to be group-size dependent, as discussed above. Furthermore, our multi-model inference approach revealed that, despite more central individuals experiencing higher predation risk, being close to others inside the school was still associated with a lower risk of being targeted.

As most prey experience many types of predators, including sit-and-wait predators and active predators that hunt for prey, the extent and direction of such selection effects will depend on the broader predation landscape in which prey find themselves. While the finding that pike were more likely to attack the main school may also appear to indicate a selective disadvantage to school, calculating the per-capita-risk for each individual would actually reveal it is still safest to be part of the main school. Nevertheless, as the shiners in our study rarely exhibited fission-fusion dynamics we feel our dataset is not appropriate to make proper inferences about how predation risk is linked to group size.

Laboratory studies on predator-prey dynamics like ours do, of course, have their limitations. Although the size of the arena we used (~2m²) is in line with behavioural studies with large schools of fish (e.g. *Sosna et al., 2019*; *Strandburg-Peshkin et al., 2013*) and experiments with live predators attacking schooling prey (*Bumann et al., 1997*; *Magurran and Pitcher, 1987*; *Neill and Cullen, 1974*; *Romenskyy et al., 2020*; *Theodorakis, 1989*), compared to conditions in the wild the prey and predator had limited space to move. However, as pike are ambush predators they tend to move relatively little to search for prey and rather rely on prey movement for encounters (*Nilsson and Eklöv, 2008*). Increasing tank size would have made effective tracking extremely difficult, or impossible, and while a much larger tank is expected to considerably increase latency to attack, we expect it to have relatively little effect on the observed findings. This was primarily done to be able to keep track of individual identities and compute features related to the visual field of the fish. Shiners naturally school in very shallow water conditions as well as near the surface in deeper water in the wild (*Hall et al., 1979*; *Krause et al., 2000b*; *Stone et al., 2016*) and also pike primarily occur in the shallow littoral zone, sometimes only a few of tens of cm deep (*Pierce et al., 2013*; *Skov et al., 2018*). Furthermore, pilot experiment showed the pike did exhibit normal swimming and attack behaviour with attack speeds and acceleration comparable to previous work (*Domenici and Blake, 1997*; *Walker et al., 2005*). Recent other work on predator-prey dynamics did not find a considerable impact of adding the third dimension to their analyses (*Romenskyy et al., 2020*). Still, the water depth used is a limiting factor of our study and in the future this type of work should be extended to deeper water while still keeping track of individual identities over time. We expect that adding the third dimension would not change the stealthy attack behaviour of the pike and therefore still put more central individuals most at risk, but possibly attack success would be reduced because of increased predator visibility and prey escape potential in the vertical plane, which remains to be tested.

For our experiments, we used a testing arena without any internal structures such as refuges. This was a strategic decision as providing a more complex environment would have impacted the ability of the shiners to school in large groups and would have led fish to hide under cover. Although studying predator-prey dynamics in more complex environments would be interesting in its own regard, it would not have allowed us to study the questions we are interested in about the predation risk of free-schooling prey. Furthermore, pilot experiments indicated that the pike never used refuges (consistent with previous work, see *Turesson and Bronmark, 2004*), so they were not further provided during the actual experiment. Experiments were conducted under artificial lights with reduces intensity but at a level high enough to be able to acquire accurate videos of the trials without motion blur. In the wild, pike are however generally found to be most active during dusk and dawn (*Raat, 1988*; *Skov et al., 2018*) and to consume most prey at low light intensity (*Dobler, 1977*). We expect that if a lower light intensity was used, the pike may profit from visual superiority and thereby would have increased predation success, further aided by a likely loosening of the prey schools due to limited light being available (*Dobler, 1977*). While it is now increasingly possible to obtain detailed data from predation events on grouping prey in the field (see e.g. *Handegard et al., 2012*; *Krause et al., 2017*), even with the most sophisticated field-based imaging it would not have been possible to acquire the highly detailed data we obtained here. That is, individual-level characteristics of predator and all grouping prey throughout predator attacks at high spatial and temporal resolution, linked to attack success and survival. Future work is now needed that further considers the different relevant ecological factors, for example deeper water, more heterogeneous environments with vegetation, different light levels, and how they interplay with the observed effects of differential predation risk in schools of fish.

In conclusion, using a quantitative empirical approach in which we acquired highly detailed individual-based characteristics of predators attacking grouping prey, we provide key mechanistic insights into when and where predators attack coordinated mobile groups and what predicts individual predation risk and survival. We demonstrate that ambush predators such as pike can stealthily approach groups

and delay their attack until being very close to, and often even until being surrounded by their prey. We also show that, rather than just go for the group centre, it pays for animals to position themselves near others, align with their body orientation, and avoid being positioned close to, and in front of, the predator. Even central individuals, when targeted, have a chance to escape an attack by avoiding the predator's head and strongly accelerating in response to the attack. Our study provides key insights about differential predation risk in groups of prey and highlights a fundamental role for both predator attack strategy and decision-making and prey behaviour. This may have important repercussions for the costs and benefits of grouping and thereby the distribution of (social) phenotypes in populations, and the collective properties of animal groups (*Farine et al., 2015*; *Herbert-Read et al., 2017*; *Jolles et al., 2020*). It is therefore paramount for future work to consider the multi-faceted features of both predator and prey and the role of the predator in the broader predation landscape to properly understand its role in the evolution of animal grouping.

## Materials and methods
### Study species and animal holding
For the design and execution of our study we made sure to adhere to the guidelines of the STRANGE framework (*Webster and Rutz, 2020*) and the standards set forth by the *ASAB/ABS, 2012* and the guidelines for predation experiments described by *Huntingford, 1984*. We used golden shiners (*Notemigonus crysoleucas*) as prey and Northern pike (*Esox lucius*), a common predator of shiners (*Johannes et al., 1989*; *Nursall, 1973*), as predator in our experiments. Juvenile shiners and pike were respectively purchased from Anderson Farms in Lonoke, Arkansas, USA and the New Jersey Division of Fish and Wildlife hatchery. Fish were reared communally in semi-natural conditions at the hatcheries and fed a diet of pelleted food. After arriving at the Princeton University laboratories, fish were kept under controlled conditions (water temperature 16.5°C ± 1°C; room lighting: 12:12 hr light:dark cycle), with shiners and pike kept in separate rooms under social conditions reflecting their natural social context. Shiners were housed in groups with 20 individuals in 37 L glass tanks on a flow-through system and fed pelleted food (Zeigler Finfish) ad libitum once daily except for the day prior to an experimental trial. Pike were housed individually in 114 L tanks containing gravel and artificial plants, and were fed a single shiner every other day for the first 10 days after arrival after which they were only able to feed during the experimental trials (see details below). All pike readily took and ate the shiners in their home tank. Experiments started after two weeks of acclimation in the laboratory. We ran two batches of experiments with the shiners within a batch all being of similar age and size (batch 1: 7.9 cm, 95% CI: 6.7–9.8; batch 2: 10.0 cm, 95% CI: 8.6–11.9 cm) and the pike being about 2.5 x their size (batch 1: 20.0±0.1 cm, n=9; pike batch 2: 27.9±0.4 cm, n=4). The sex of the fish was not determined. As shiners mainly school when they are juveniles, sex is not expected to play a major role in the observed findings. No instances of poor health or body condition were observed among any of the experimental fish throughout the experimental period.

### Experimental arena
Trials took place in a 3.5 × 6.5 ft (1.06 × 1.98 m) white Perspex tank. External disturbances were minimised by placing the tank on two layers of carpet (to dampen vibrations), surrounded by white curtains, and positioned it in an otherwise empty experimental room. Diffused light was provided by two LED panels positioned outside the curtains at the far ends of the tank. The tank was filled with water to a depth of 6 cm, about 1.5x – 2x the body height of the pike, that had approximately the same temperature and quality as the water of fishes' home tanks. The right-bottom corner of the tank contained an opaque Plexiglass partition to visually separate the pike from the shiners at the start of each trial. A Sony NEX-FS700 camera positioned at 2 m above the exact centre of the tank was used to film the experimental trials at 120 fps with a resolution of 1920 × 1080 pixels.

### Experimental procedure
Before the first trial, each pike was acclimated to the experimental arena by two mock trials with one and subsequently three shiners on two separate days, while each shiner group was allowed to acclimate to the experimental tank for 24 hr before their first pike exposure. An experimental trial started with netting a pike from the nearby holding room, transferring it to the experimental room in

a covered bucket, and immediately releasing it into the partitioned corner of the experimental tank. While the pike was left to acclimate in the holding corner, a group of 40 shiners was taken from the separate holding room, transferred to the experimental room in a covered bucket, and released into the centre of the experimental tank. Subsequently, the experimenter closed the curtains around the tank, started video recording, and moved to a separate isolated section of the room. Shiners were allowed to acclimate to the tank for five minutes. After that, the partition was raised using a remote pulley system, giving the pike access to the whole tank. Ten min later video recording was stopped and, to decrease potential stress, experimental lights were turned off and dim red lights turned on. To keep the experimental period consistent across all trials, trials were not stopped after the first shiner was captured or eaten. About a minute after turning on the red lights the shiners were carefully netted from the experimental tank and returned to the holding tanks and immediately fed with pelleted food. Subsequently, the pike was netted from the experimental arena in a gentle way and transferred to its home tank. After each trial, we drained and scrubbed the experimental tank thoroughly with a sponge to remove any potential predator and alarm cue odours. Three to four experimental trials were run per day between 10:00 and 18:30.

We ran the experiment with two batches of pike and shiners, with the fish of each batch tested repeatedly in the arena over time. Pike were randomly selected, but only after they had at least three rest days since the previous trial, which was decided based on pilot work that showed pike were less motivated when tested with less time between trials. For each trial, shiners were also randomly selected from the holding tanks to create groups of 40 fish. Shiners were mixed to avoid potential effects of familiarity between individuals and the associated social feedback and learning effects related to group composition. However, as we used a repeated-measures design, we made sure that all shiners in the same group had the same number of exposures to the pike by keeping shiners in separate social holding tanks based on their number of pike exposures. In total we used 20 pike and started with about 1500 shiners. Pike had a mean number of 1.84±0.14 attacks per trial, with 18 trials having more than 1 successful attack. However, not all pike did always attack during the trials, and 7 pike never attacked. As those trials did not provide any data on the attacks they were excluded and the pike not used in further trials. Also the shiners were excluded for subsequent trials as their future behaviour in the assay could be influenced by having had experience with a pike that did not attack. As a result, our sample size decreased with exposure and we stopped at a maximum number of 6 exposures. Like the pike, shiners were also generally tested with three rest days between trials but sometimes groups had one or a few fish that only had two rest days. This was inevitable due to the difficulty of just being able to run a few trials each day, and sometimes pike thus not attacking. Although none of the fish had any previous experience with any experiments, during the repeated exposures both shiners and pike had time to learn about the conditions they were confronted with, thus enabling us to also investigate how this impacted predators' attack behaviour and prey response.

## Fish tracking

After automatically correcting all videos for minor camera lens distortion, we used custom developed tracking software to acquire highly detailed individual-based movement data for both the shiners (SchoolTracker, by Haishan Wu) and the pike (ATracker, by J. W. Jolles), including head position and body orientation in two dimensions. Full details of how SchoolTracker detects fish, tracks their movement, and corrects occlusions can be found in *Rosenthal et al., 2015* ATracker uses background subtraction algorithms and blob detection, shape, and threshold algorithms to get the pikes' position and orientation while excluding the shiners. For our analyses we manually checked for constancy of individual shiners' identities in the large schools at 120 fps from one second before strike initiation through to one second later, and manually corrected identity swaps and centroid and heading positions where needed. The identity of the targeted individual was acquired by carefully comparing the video and visualisations of the tracking data at the moment of the attack. For the far majority of unsuccessful attacks we also acquired the identity of the targeted shiner with high certainty because the pike attacked the prey from very near and exhibited a clear attack trajectory. In the few cases, there was a second potential target we were able to determine the most likely target by carefully going back and forth through the video, frame-by-frame, at the moment of the attack.

## Quantification of behaviour

After tracking, we smoothed the positional coordinates using a Savitzky-Golay smoothing filter with a window of 0.1 s and converted pixels to mm. Using the head as reference, we then computed each fish's velocity, speed, and acceleration as well as their distance to the closest wall. For each shiner we also computed their distance to the pike, their angle to the pike (absolute, with 0° being directly in front of and 180° directly behind the pike), and relative orientation to the pike (with 0° being the same orientation as the pike and 180° an opposite orientation). We did this by shifting the coordinates of the fish so that the origin of the coordinate system was at the pike and rotated such that the pike was pointed north (0°) to have a common frame of reference. The maximum speed and acceleration values we report in the text are based on the smoothed data, which helps overcome the issue that already tiny shifts in reference points in subsequent frames during tracking can confound single data values. Computing these metrics on unsmoothed data instead results in maximum speed and acceleration values that are in line with that typically reported for attacks or escape (*Domenici and Blake, 1997*; *Walker et al., 2005*).

Next, we used a hierarchical clustering approach to determine the distribution of shiners in one or multiple groups. In short, we computed clusters hierarchically by starting with each shiner assigned to its own cluster and iteratively joined the two most similar clusters based on the centroids of those clusters until there was just a single cluster. The optimal number of clusters was determined automatically based on the change in distance between the clusters' centroids in a single step relative to the variance in cluster distance. We thereby looked for large discontinuities in the change in cluster distance and used a predetermined threshold to optimize the clustering. This approach provides more realistic clustering than other approaches, such as those that use a fixed distance measure for clustering. For all attacks, we made sure to manually check the computed clusters at the time of attack, and in a few cases made a correction when there was a large disparity in body orientation (>90°) between a fish and the rest of the group. For each attack, we then ranked the clusters and focused subsequent analyses on all attacks where pike attacked the largest cluster (n=117). To get a realistic representation of the boundaries of the group (i.e. the largest cluster), we ran convex and subsequently concave hull approximations based on the position of all shiners in the group and scored if fish were on the group boundary as well as pikes' distance to the group hull, with negative values indicates the pike was inside the group boundaries.

Next, for each shiner we quantified their Voronoi polygon area, limited domain of danger (LDOD; following *Lambert et al., 2021*), local misalignment, and their inter-individual distance (see *Table 1* for an explanation how these measures were computed). LDOD and local misalignment require a spatial threshold outside of which fish will be included. We considered the distance within which fish on average had 25% of their neighbours, which provides a good balance between generating enough variation among individuals and the area containing too many individuals. As across all trials fish on average had 25% of their neighbours within a distance of 9.9 cm, we used a threshold of 10 cm to include neighbouring fish for our LDOD and local misalignment measures. At the group level, at each time point throughout the attacks we determined the group's (i.e. the largest cluster) position and movement vector based on its centre of mass and calculated group speed, cohesion, the average inter-individual distance between all shiners, polarization, which is a measure of the alignment of the fish in the group relative to each other and ranges from 0 (complete non-alignment) to 1 (complete alignment), and milling, which is a measure of high local but low global alignment as the group rotates around its core (*Tunstrøm et al., 2013*). Subsequently, we computed shiner's absolute distance to the group centroid, and their distance along the front-to-back axis (positive when ahead of the centroid, negative when behind). As for the individual shiners, we computed the groups' position, heading, and orientation relative to that of the pike. To make sure group relative heading changes were not due to changes in pike position, we calculated the change in position of the group relative to the pike from 0.1 s before strike initiation while keeping the pike's position constant. We thereby considered the same fish such that group centroid position could not fluctuate due to potential changes in cluster size. This movement of the group centroid equates to the average direction of motion of the prey fish.

Finally, to estimate the visual information available to each fish we used a ray-casting algorithm, originally developed for *Rosenthal et al., 2015* (for further details see their paper). Visual features computed using this method have been shown to be informative of evasion behaviour (*Rosenthal et al., 2015*; *Sosna et al., 2019*), even in field conditions (*Hein et al., 2018*). We used the visual

information to compute the proportion of each shiner's vision that was occupied by conspecifics (weighted degree) and the proportion of a shiner that was visible to the pike. While individuals form a relatively planar group structure near the water surface, as schools are not perfectly two-dimensional, it may be the case that neighbouring individuals do not always block an external view. However, shiners tend to form relatively planar groups near the water surface (*Hall et al., 1979*; *Stone et al., 2016*), and using incomplete rather than full blockage seems to have very little effect on an individual shiner's detection coverage (*Davidson et al., 2021*).

## Analyses

To investigate if pike had a higher maximum speed than the shiner they targeted, we computed both fishes' (smoothed) speed from 0.5 s before until 0.1 s after strike initiation and ran a linear mixed model with fish as fixed factor (predator, prey), maximum speed as response variable, and attack id as a random factor. To determine if groups were more likely to attack the groups head-on in terms of their orientation and direction of motion/heading, that is if their angles concentrated around 180° relative to the pike pointing north (0°), we ran Rayleigh tests for uniformity, with the Rayleigh statistic varying from $R=0$, indicating a uniform distribution in all directions, to $R=1$, representing that all vectors point in the same direction at 180°. For the analysis regarding the group heading, we made sure to subset the data to attacks where the group was moving at least at a speed of 1.5 cm/s (n=102). To investigate how repeated exposure changed the shiners packing fraction and avoidance of the pike's head and pikes' distance from the targeted individual, we ran a linear model with exposure, fitted as a cubic function, as a fixed factor and data subsetted to pikes' first attack attempt during the trials. To determine what features were most predictive of whether a shiner is targeted by the pike and survives an attack, we used a multi-model inference approach (*Burnham et al., 2011*), focusing our analyses on all attacks where the pike attacked the main school (n=117), described in detail in Appendix 3. Thereby estimates, likelihood-ratio tests, and p-values reported in the text were acquired from the final models. In some cases, we included Pearson correlations and linear models to further investigate the relationship and explained variance ($R^2$) between two predictor variables.

## Acknowledgements

We thank the New Jersey Division of Fish and Wildlife for providing the pike used in this study and for their valuable advice, three anonymous reviewers for their helpful comments and suggestions, and Mike Gil and Jake Graving for constructive feedback on a previous version of the manuscript. We acknowledge support from the Alexander von Humboldt Foundation (Postdoctoral Fellowship to J.W.J), the Zukunftskolleg, Institute for Advanced Study at the University of Konstanz (Postdoctoral Fellowship to J.W.J), the Severo Ochoa Program for Centres of Excellence funded by the MCIN and the AEI (Postdoctoral Grant to J.W.J, #CEX-2018–000828 S), NSF-DDIG (Graduate Research Fellowship to M.M.G.S, #1701289), MindCORE (Postdoctoral Research Fellowship to C.R.T), the Center for an Informed Public and the John S and James L Knight Foundation (support to J.B.-C.), the Office of Naval Research (Research Grant to I.D.C, #N00014-64019-1-2556), the European Union's Horizon 2020 research and innovation programme under the Marie Skłodowska-Curie grant agreement (to I.D.C; #860949), the Struktur- und Innovationsfonds für die Forschung (SI-BW) of the State of Baden-Württemberg, the Deutsche Forschungsgemeinschaft (DFG, German Research Foundation) under Germany's Excellence Strategy-EXC 2117–422037984 (to I.D.C.), and the Max Planck Society.

## Additional information

### Funding

| Funder | Grant reference number | Author |
| --- | --- | --- |
| Alexander von Humboldt-Stiftung | | Jolle Wolter Jolles |
| Ministerio de Ciencia e Innovación | CEX-2018-000828-S | Jolle Wolter Jolles |

| Funder | Grant reference number | Author |
|---|---|---|
| National Science Foundation | 1701289 | Matthew MG Sosna |
| Universität Konstanz | | Jolle Wolter Jolles |
| John S. and James L. Knight Foundation | | Joseph Bak-Coleman |
| Office of Naval Research Global | N00014-64019-1-2556 | Iain D Couzin |
| HORIZON EUROPE Marie Sklodowska-Curie Actions | 860949 | Iain D Couzin |
| Deutsche Forschungsgemeinschaft | EXC 2117-422037984 | Iain D Couzin |

The funders had no role in study design, data collection and interpretation, or the decision to submit the work for publication.

### Author contributions

Jolle Wolter Jolles, Conceptualization, Data curation, Software, Formal analysis, Supervision, Validation, Investigation, Visualization, Methodology, Writing – original draft, Writing – review and editing; Matthew MG Sosna, Conceptualization, Resources, Formal analysis, Funding acquisition, Validation, Investigation, Visualization, Methodology, Project administration, Writing – review and editing; Geoffrey PF Mazué, Investigation, Writing – review and editing; Colin R Twomey, Software, Writing – review and editing; Joseph Bak-Coleman, Methodology, Writing – review and editing; Daniel I Rubenstein, Resources, Supervision, Writing – review and editing; Iain D Couzin, Conceptualization, Resources, Supervision, Funding acquisition, Methodology, Writing – review and editing

### Author ORCIDs

Jolle Wolter Jolles (iD) http://orcid.org/0000-0001-9905-2633
Iain D Couzin (iD) http://orcid.org/0000-0001-8556-4558

### Ethics

This study was performed in strict accordance with the standards set forth by the ASAB/ABS Guidelines for the Treatment of Animals in Behavioural Research (2012) and the guidelines for predation experiments described by Huntingford (1984). Specifically, staged predation events, whereby live predators could interact freely with and consume their prey, were necessary to quantify normal predatory and anti-predator behavior as well as individual fitness and thereby realise the novel objectives of our study, going beyond previous work using predator cues or models or with virtual prey. We thereby acquired highly detailed data of all attacks, something that would not have been possible in the wild and with the aim to get the maximum possible information from each trial (c.f. Huntingford, 1984). We were able to reduce the number of fish used in the experiments by conducting repeated exposures, combining biological (different groups) and technical (independent repeated measures) replicates. Although shiners may experience stress during the staged predation encounters, the testing conditions with a group size of 40 fish, which reflects the size of shiner shoals observed in the wild (Hall et al., 1979; Krause et al., 2000), and the large open tank, enable shiners to hide among others and escape attacks. All animal care and experimental procedures were approved by the institutional animal care and use committee (IACUC) protocols (#2068-16) of Princeton University.

### Decision letter and Author response

Decision letter https://doi.org/10.7554/eLife.76344.sa1
Author response https://doi.org/10.7554/eLife.76344.sa2

## Additional files

### Supplementary files
• MDAR checklist

## Data availability

Associated datasets are available on Mendeley Data (https://doi.org/10.17632/bszk9ztryp.1).

The following dataset was generated:

| Author(s) | Year | Dataset title | Dataset URL | Database and Identifier |
|---|---|---|---|---|
| Jolle J, Matt S | 2021 | Data for: Both Prey and Predator Features Determine Predation Risk and Survival of Schooling Prey | http://doi.org/10.17632/bszk9ztryp.1 | Mendeley Data, 10.17632/bszk9ztryp.1 |

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

## Appendix 1.

## Quantification of the time of attack

Pike attacks are characterized by a sharp, sudden, burst of movement, the 'strike'. To automatically determine the exact frame of strike initiation ('time of attack'), we quantified the work rate by looking at the change in kinetic energy c.f. (*Sosna et al., 2019*). In short, for an individual with mass $m$ and speed $s$, the change in kinetic energy is given by: $\Delta KE_i = ms_i \frac{ds_i}{dt}$ . As here we simply use kinetic energy to determine the frame of strike initiation, we considered the mass as a constant across all pike. We then rescaled this measure by $1/m$ to get "swimming intensity". To automatically differentiate strikes from normal swimming, we focused on the peak intensity an individual attained and compared that to a swimming intensity threshold of two standard deviations above its median intensity in the second (120 frames) prior to its peak intensity during an attack. Normal swimming lacks a simultaneous high speed and acceleration, thus enabling us to automatically and objectively determine the exact time of attack at 120 fps (see *Appendix 1—figure 1*), which was confirmed by manual video inspections.

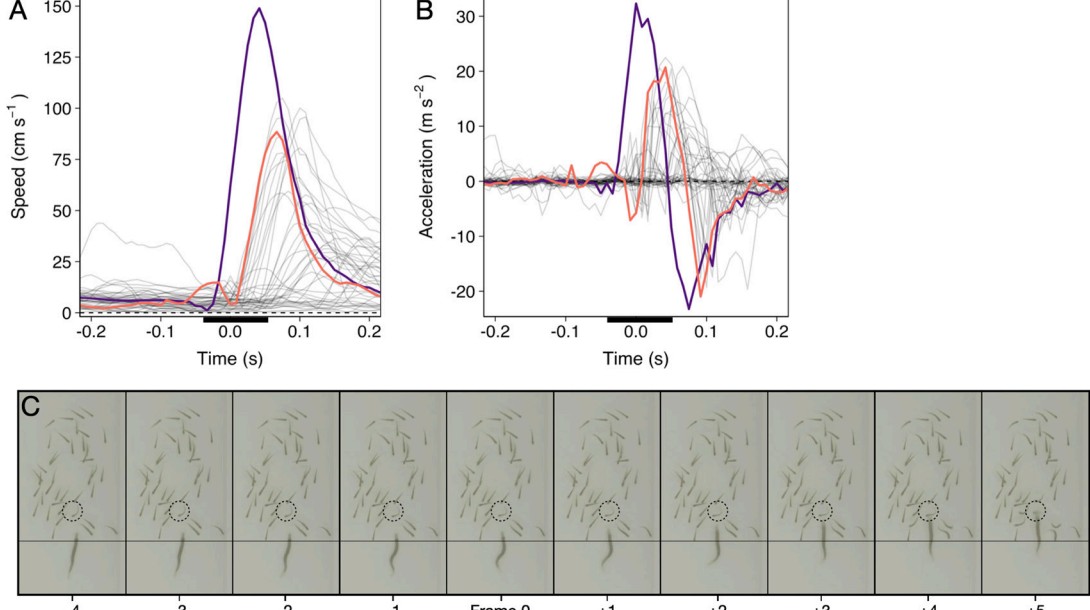

**Appendix 1—figure 1.** Time series of a randomly selected pike attack. (**A and B**) Speed and acceleration curves (based on smoothed data) of the pike (purple) and shiners (grey; targeted shiner in orange). Time is relative to the automatically quantified time of attack. Black bar reflects time series of panel C. (**C**) Cropped screenshots of the frames around the strike (indicated by the black bar in panels A and B), showing the characteristic S-shaped body posture leading up to the strike, during which the pike's head stays roughly at the same position (see thin horizontal line). Targeted individual is indicated by the dashed circle. This specific strike lasted a total of 6 frames, or 0.05 s, until impact.

## Appendix 2

### Effect of repeated exposure

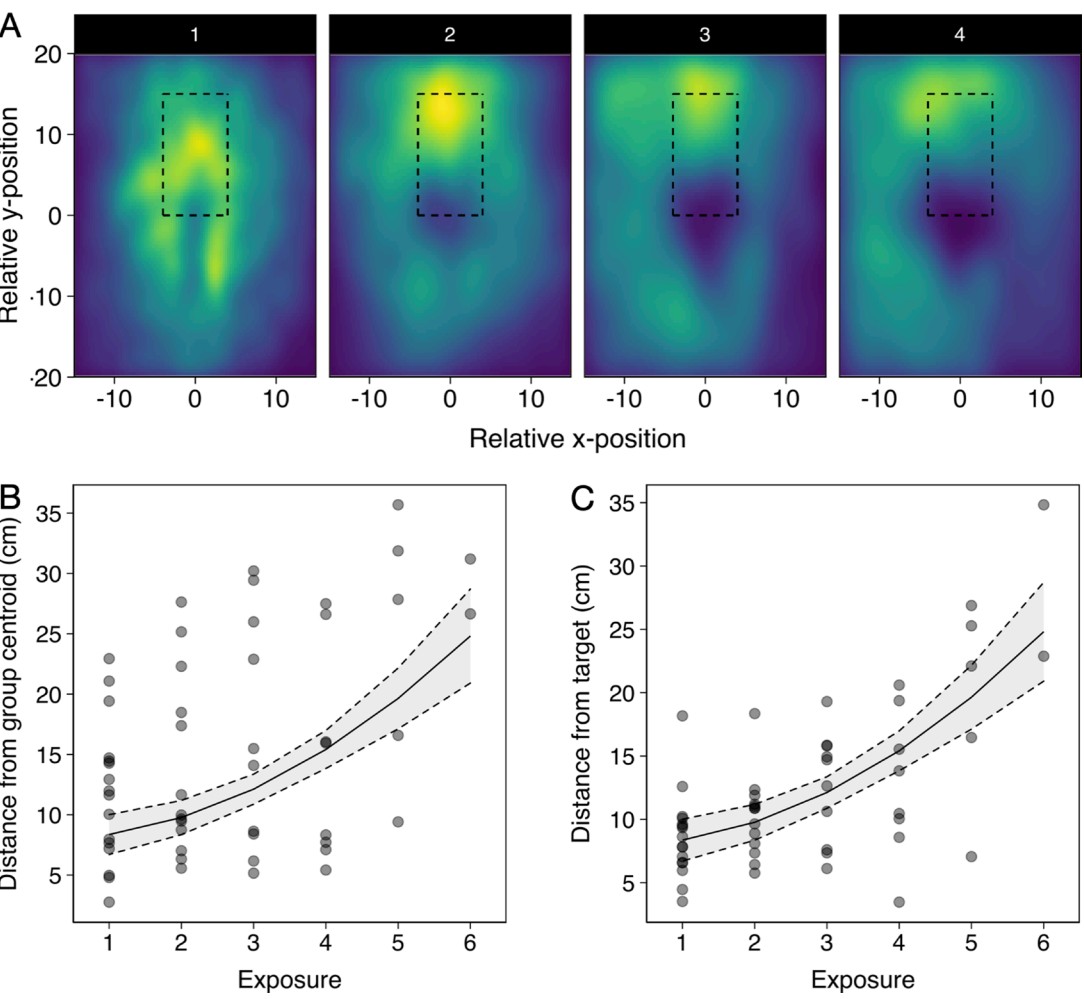

**Appendix 2—figure 1.** Effects of repeated pike exposure on relative spatial positioning of the shiners and pike. (A) Heatmaps of the relative position of all shiners to the pike up to strike for all first attempts relative to exposure, with the 4th-6th exposure (panel 4) being grouped due to their smaller sample size, and data subsetted to the relevant region around the pike (nr of datapoints, that is a shiner location at a given frame, per attempt: 1=650.000; 2=528.950; 3=325.570; 4+=319.995). Rectangular area shows the area in which all attacks occurred. (B and C) Plots of the relationship between pikes' distance to the group centroid (B) and to the targeted individual (C) at 0.5 s before the strike with repeated exposure. Semi-transparent points indicate individual trials. Line and 95% confidence intervals (dashed lines) are from a linear model with exposure fitted as a cubic function. Data was subsetted to pikes' first attack attempt during the trials.

## Appendix 3

### Feature selection for predictors which shiners are targeted and survive attacks

To determine what features are predictive of whether a shiner (i) is targeted by the pike and (ii) survives an attack, we used multi-model inference (*Burnham et al., 2011*) focusing our analyses on all attacks where the pike attacked the main school (n=117). In multi-model inference, rather than fitting a single model, a model is fit for every possible combination of features (predictor variables) and the relative importance of each feature can be assessed based on which features are present in the best models. We included a large range of potentially relevant features to characterise the prey and thereby determine their likelihood to be attacked (*Appendix 3—figure 1*). We then ran exhaustive (i.e. not stepwise) model evaluation of all possible feature subsets to determine optimal feature importance (*Johnson and Omland, 2004*). Models were weighted based on their BIC score, which places a larger penalty on regression coefficients than AIC to avoid overfitting, after which relative feature importance was calculated based on all models that contained the feature, following *Burnham et al., 2011*; *Rosenthal et al., 2015*. We thereby excluded any models that contained both absolute and ranked variables of the same feature (e.g. distance to centre and ranked centre-to-edge position) as these were inherently strongly correlated. We also made sure other included features did not show high collinearity by checking their Variance Inflation Factor (using a VIF criterion of 5, *Harrison et al., 2018*). The final predictive features were selected based on their relative importance score across all models as well as by considering the features that appeared in the most parsimonious models (see *Appendix 3—figures 1 and 2*).

We started with a prey-focused approach to investigate predation risk purely from the prey's perspective and thereby considered all shiners that were part of the main school to be potential targets for the pike. We ran generalized linear models with a logistic link function, with as response variable whether an individual was targeted or not (1,0). To account for the hierarchical structure of our data, we included as random effects exposure, Pike ID and attack attempt (1st, 2nd, 3rd) nested within trial. As potential predictive features we included distance to the group centroid (cm), front-back centroid distance (cm), Voronoi area, Voronoi area limited to a maximum radius of 10 cm from a conspecific (LDOD), median inter-individual distance (cm), weighted degree, and whether an individual was positioned on the group boundary (yes/no). We also included ranked variables for distance to group centroid and front-back centroid distance, further scaled between 0 and 1 to account for variation in cluster size, which better reflect individuals' relative positioning in the group. To investigate predation risk from the predator's perspective, we also considered features that included information about the pike, specifically shiners' distance, angle, and relative orientation to the pike, and how much the pike saw of each shiner. Furthermore, we now only considered shiners within the attack region, the region in which all attacks occurred, 8 cm wide and 15 cm long directly in front of the pike. To quantify what best predicted an individual's likelihood to survive an attack, we used predation success as our (binary) response variable and again ran multi-model inference, considering all significant features that arose from our prey- and predator-focused approaches looking at which individual was targeted. We additionally included pikes' maximum acceleration in the 0.5 s prior to strike initiation, shiners' maximum speed, acceleration, and turning angle in the 0.5 s prior to strike initiation, and pike's vision of the shiner. As targeted shiners' maximum speed and acceleration were positively linked ($r$=0.68), all models that contained both these features were excluded. Furthermore, measures of maximum speed and acceleration throughout the full attack were not included as these could be confounded by the difference in duration of the attacks observed. How much the targeted shiner saw of the pike was not included either because this measure was found to be strongly linked to pike's vision of the shiner ($r$=0.40).

Our measures of local misalignment and LDOD were quantified for all neighbouring fish within a radius of 10 cm as this on average equated 25% of the school. To make sure the range over which we investigated local misalignment did not influence its predictive power, we also ran a model for the prey-focused approach with local alignment computed for individuals within a topological neighbourhood of 6 neighbours (c.f. *Ballerini et al., 2008*). Although this slightly weakened model fit ($\Delta$BIC = +8.9) it did not qualitatively change the model results. Similarly, as front-back positioning is not meaningful for schools that are not well aligned, we re-ran model selection on the data subsetted to schools with a group polarization of at least 0.4 (n=80 attacks). The resulting model output was not qualitatively different from the final model of feature selection, and adding front-

back positioning as an additional feature again weakened model fit (ΔBIC = +6.3). Pike's distance to the tank walls could also be hypothesized to influence its attack success. Adding this variable to the final model did however show this was not the case (ΔBIC = +4.4).

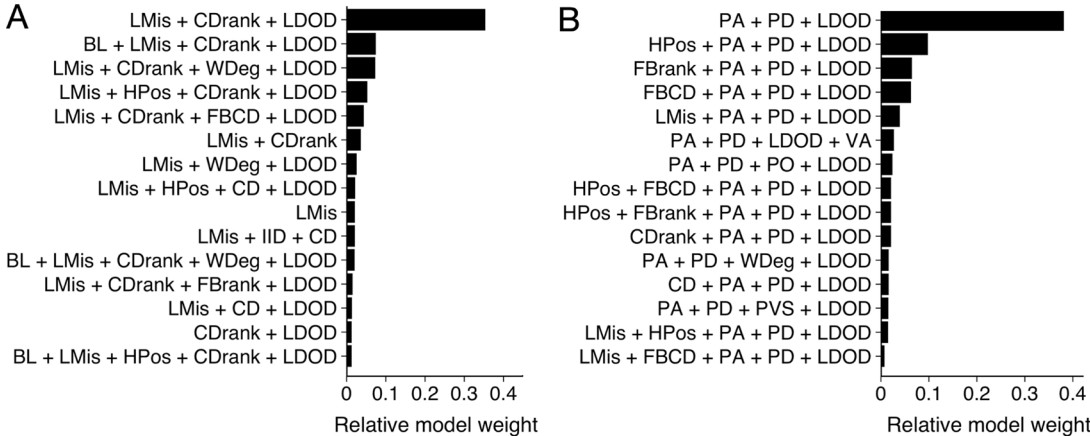

**Appendix 3—figure 1.** Model results from multi-model inference on likelihood to be targeted. Results from multi-model inference investigating which individual is targeted for attack using the prey-focused (**A**) and predator-focused (**B**) approach. Panels show relative evidence weight for the top 15 models for each approach. For acronyms, see *Table 1* in the main text.

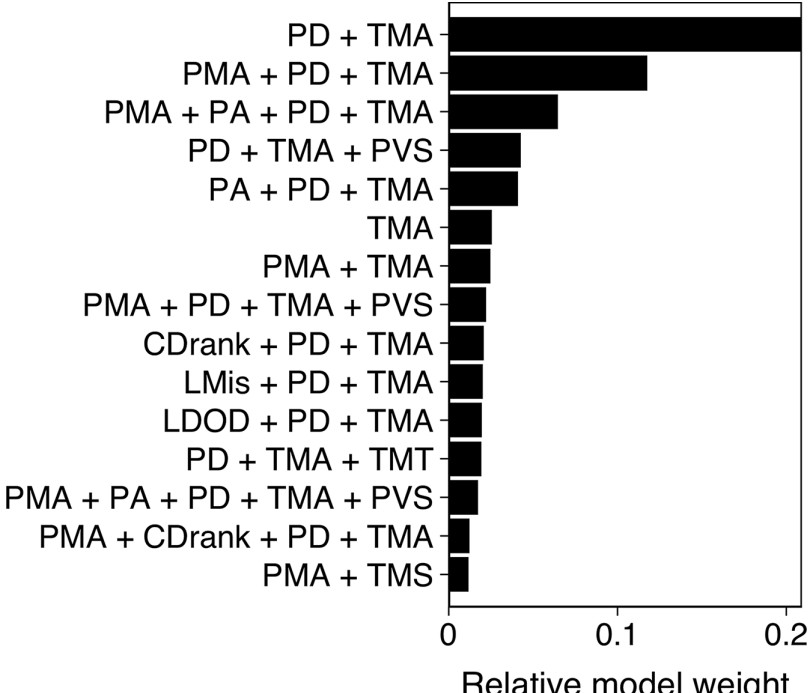

**Appendix 3—figure 2.** Model results from multi-model inference on predation attack success. Panel shows relative evidence weight for the top 15 models. For acronyms, see *Table 1* in the main text.

