## [Editor Report]

This study, which uses cutting-edge video-tracking methods to investigate predictors of predators' attack success on schooling fish, will be of interest to behavioural, evolutionary, and movement ecologists. It underscores the potential of experimental high-resolution tracking approaches and the importance of examining the perspectives of both the predator and its prey. The article is a valuable contribution to understanding predator-prey dynamics and their role in shaping the evolution of social grouping behaviour.

---

## [Decision Letter]

**Decision letter after peer review:**

Thank you for submitting your article "Both Prey and Predator Features Predict the Individual Predation Risk and Survival of Schooling Prey" for consideration by *eLife*. Your article has been reviewed by three peer reviewers, and the evaluation has been overseen by a Reviewing Editor and Christian Rutz as the Senior Editor. The reviewers have opted to remain anonymous.

The reviewers have discussed their reviews with one another, and the Reviewing Editor has drafted this decision letter to help you prepare a revised submission.

Essential revisions:

1. Please discuss the limitations of the experimental design, including limited space for the prey to move and lack of opportunity to seek refuge, as well as the possibility that the stealthy hunting strategy described for the pike is affected by tank conditions.

2. Please discuss the limitations of 2D versus 3D tracking. Although the manuscript includes some arguments for why 2D tracking is sufficient, one of the reviewers noted that the primary supporting citation is for a prey-focused study design for different predator and prey species. Rather than making the assumption that the third dimension does not matter, please evaluate critically how violations of that assumption might affect your inferences/conclusions (or motivate further work) and ensure that this discussion is incorporated in the main text to make it apparent to readers.

3. Please clarify how the present results contribute to our understanding of the evolution of collective behavior in the wild, taking into account the potentially counterintuitive observation that fish in smaller clusters experienced lower predation. And please address Reviewer #2's related question about whether repeated exposure leads to reduced schooling behavior overall.

4. Please clarify all questions related to experimental design, animal numbers, the outcome of prey attacks, and statistical analyses, and add an explanatory figure, as suggested by Reviewer #3.

*Reviewer #1 (Recommendations for the authors):*

Overall this paper is beautifully written and presented, with appropriate analyses.

It is packed full of information, and as a result, it can be a little dense and hard to follow at times. Some streamlining, signposting and highlighting of key points would be of benefit.

One solution may be clearer hypotheses or at least clearer aims. This would add some further structure to the text. As a reader, going from the Introduction to the first line in the Results section being about describing pike movements, it was a little tricky to immediately follow the logic and flow.

From reading everything, it is still not clear to me if the pike actually caught and killed the shiners? It feels a little like the ethics section dances somewhat around this point. Please clarify.

The authors allude to the captivity aspect in their Discussion but I think this needs to be greater. There is still limited space for the prey to move and react. You mention this, but you don't expand on what you think would happen if the prey had endless options?

Do pike ambush easier when extensive cover is provided?

In the Introduction you state: "This not only has crucial implications for the selection of individual phenotypes, but also for the emergence of collective behaviour, and the dynamics and functioning of animal groups (Farine et al., 2015; Jolles et al.,2020; Ward and Webster, 2016). Hence, better understanding what factors influence individual predation risk within animal groups is of fundamental importance." I think this needs to be clearer. Why is it actually of fundamental importance? What real benefit is there to finding this out, beyond modelling something? Linked to this is a frequent reference to the same work, and/or the same authors. This started to feel a little like an echo chamber. I would suggest some diversification in the literature you refer too.

Based on the speed at which everything is happening, this can change in an instant. What time period was this considered over? Apologies if I missed this. "As most predators only have a specific region that they are biologically capable of attacking, we also only considered shiners found inside the pikes' strike zone, an area of roughly 8 cm wide and 15 cm long directly in front of the pike within which all targeted prey were positioned (Figure 2E)."

Based on Figure 1, how relevant is the centroid?

How much do the lighting conditions (1) influence how pike hunt, and (2) determine how the shiners behave in response to a threat?

"The behavioural type of the shiners and pike was not determined but likely showed a natural range". This seems very vague and somewhat assumptive. I think there's plenty of evidence to suggest this could go either way? Please provide a little more detail here.

"…have been of long-standing interest (Ioannou et al., 2012; Krause and Ruxton, 2002; Ward and Webster, 2016)". I'm not convinced you can say long-standing and then present references which, in the big scheme of things, are all relatively recent.

"However, besides by predators using a stealthy approach as we show here, central individuals may also be more at risk by predators". Very clunky sentence that I had to read a few times to understand what you meant. It might benefit from a minor re-write.

*Reviewer #2 (Recommendations for the authors):*

1. The second paragraph in the introduction came off as more of a list of conflicting evidence rather than explaining a topic. Perhaps it could be restructured to better motivate the following paragraphs by stating something like "Previous work has identified many different, and sometimes contradictory, factors that predict prey vulnerability in a collective group, likely because predation strategy has been overlooked."

2. What was the top speed of the shiners? This might be relevant to report in line 145.

3. In line 170, it might also be useful to refer to the distances between the pike and the centroid of the school in terms of the percent of total size.

4. Line 142: Consider changing “compromised” to “comprised”.

5. Line 337: Consider changing “However, besides by predators” to “In addition to predators”.

6. The topics of the paragraphs starting on lines 346 and 419 are overlapping. Perhaps these paragraphs can be combined so that the results of the study and the previous work in the field can be discussed together?

7. The sentence starting on line 434 is very long and confusing. Consider rewording.

8. The sentence starting on line 452 is also very long and unfocused. Consider rewording.

9. Consider breaking up the sentence starting on line 468: “…the predator. Even central individuals have a cha”ce".

10. Line 575: What does "we assured perfect tracks" mean?

*Reviewer #3 (Recommendations for the authors):*

Overall, I enjoyed reading this impressive paper. It was well written and provided a nice example of the modern, highly quantitative analyses of predator-prey attack dynamics in controlled laboratory conditions. I had only a few relatively minor comments.

The authors did highly detailed analyses of metrics that explained which prey individual was targeted by the predator, and whether the targeted individual successfully evaded capture or not. This assumes that it is clear cut that there was a particular individual targeted and that the analysis correctly identified the 'targeted' individual. Given that there were probably some, if not many, attacks where multiple prey were directly in front of the predator and close by, what exact criteria did the authors use to identify which prey was targeted by the predator? In the 70% of attacks where a prey fish was captured, it is reasonably clear that that was the prey that was targeted, but in the other 30%, it seems less clear cut. Were there some, perhaps quite a few, situations where it was not 100% clear which prey was targeted? Conventional wisdom includes the idea that attack success is reduced if predators get 'confused' by having numerous prey that are potential targets, particularly if they escape in different directions. If the authors mis-identified which prey was targeted this could presumably affect the accuracy of their analyses on factors that affect attack success.

Presumably to facilitate having sharp video images and 2-dimensional analyses of encounters, the trials were done in 6 cm of water which is likely much shallower than natural conditions for pike and shiners. The authors suggest that shiners are commonly in flat schools, so quantifying their position and movements in 3 dimensions is not necessary. They also cite a recent paper (Romenskyy et al. 2020) that suggested that the third dimension (variation in the vertical-depth axis) is not critically important in a similar study quantifying the details of attack, but this other study was on different predator and prey fish species, and focused only on which prey are targeted, not on attack success. Overall, the notion that it is valuable to study fish predator-prey interactions in 2 dimensions sounds plausible at a rough level, but I am somewhat skeptical of that conclusion at the very fine, detailed scale of this paper's analyses. Nonetheless, I found this study to be impressive and valuable, even if the detailed dynamics under some, perhaps many natural conditions (deeper water, with some refuge not far away, in different light conditions etc.) might be moderately different.

The experimental design and sample sizes with regard to pattern of re-use of animals across multiple days (lines 544 ff.) might be easier to understand if the authors included a summary figure. How many groups of 40 shiners began each batch? There were 9 pike in batch 1 and 4 in batch 2, does that mean that the experiment began with 360 shiners in batch 1 and 160 in batch 2? Trials continued until there were not enough fish left for another trial with that number of exposures, which I think means until there were fewer than 40 fish. But only 70% of 125 attacks were successful (about 88). If the study started with 520 prey and only 88 were killed, it sounds like the study could go through a very large number of repeated trials before it would not have enough prey to continue. I am clearly not understanding something about the design. Pike had 3 rest days between trials. Was the same rest schedule applied to shiners? Shiners were kept in tanks with fish with the same number of previous exposures, but did this mix animals with different numbers of days since their previous trial? How many shiners participated in the study? There were a total of 125 attacks, with 1.84 attacks/trial which seems to mean 68 trials, spread across 13 pike in 2 batches. But 'a number of pike' never attacked any prey. Does the 13 pike include these or are the 13 pike only the ones that did attack prey?

I will admit that I might not have outlined the paper carefully enough, but the statistical analyses described briefly in lines 193 ff., and in some detail in the Appendix involved reasonably standard model choice methods listing relative model weights, but in the main text, the authors also showed various other statistics: LRT chi-square values, F-values, with p-values, R-squares, and correlation coefficients. Perhaps I missed where they described these, but if indeed they did not do so, they should describe all statistical tests.

Model choice analyses also allow one to assign relative weights not just to each model, but to each factor by summing the weights for each model in which that variable appears. Rather than only show the top 15 models (Appendix 3), it might be useful to calculate the weights for each factor. This would complement the results and conclusions in the main text.

Perhaps I did not study the Appendices carefully enough, but the best listed models (Appendix 3 and 4) did not include number of exposures, pike ID or attack attempt, or more importantly, interactions between these random effects and the predictive features from either the prey or predator view. Appendix 2 shows that exposure number had large effects on pike distance from the group centroid and from the target individual. Did the more detailed features of targeted prey (or attack success) change with repeated exposures? For each exposure number (Appendix 2 - figure) the distances varied quite a lot. I assume that these are for the 13 different pike. Did individual pike differ consistently in their targeting tendencies, or even in their change in targeting tendencies with repeated exposure and learning? If so, this deserves more discussion.

Appendix 2, figure 1 -- what are the darker points in figures B and C?

Line 193: 'were' not 'where'.

---

## [Author Response]

Essential revisions:1. Please discuss the limitations of the experimental design, including limited space for the prey to move and lack of opportunity to seek refuge, as well as the possibility that the stealthy hunting strategy described for the pike is affected by tank conditions.

We have considerably extended our discussion and now included two dedicated paragraphs to discuss the potential limitations of our work and how the stealthy hunting strategy of the pike may be affected by these conditions. This includes a discussion of the effect of tank size (lines 472-480): “Laboratory studies on predator-prey dynamics like ours do, of course, have their limitations. Although the size of the arena we used (~2m^2^) is in line with behavioural studies with large schools of fish (e.g. Sosna et al., 2019; Strandburg-Peshkin et al., 2013) and experiments with live predators attacking schooling prey (Bumann et al., 1997; Magurran and Pitcher, 1987; Neill and Cullen, 1974; Romenskyy et al., 2020; Theodorakis, 1989), compared to conditions in the wild the prey and predator had limited space to move. However, as pike are ambush predators they tend to move relatively little to search for prey and rather rely on prey movement for encounters (Nilsson and Eklöv, 2008). Increasing tank size would have made effective tracking extremely difficult, or impossible, and while a much larger tank is expected to considerably increase latency to attack, we expect it to have relatively little effect on the observed findings.”

As well as a discussion of the lack of refuges (lines 495-502): “For our experiments we used a testing arena without any internal structures such as refuges. This was a strategic decision as providing a more complex environment would have impacted the ability of the shiners to school in large groups and would have led fish to hide under cover. Although studying predator-prey dynamics in more complex environments would be interesting in its own regard, it would not have allowed us to study the questions we are interested in about the predation risk of free-schooling prey. Furthermore, pilot experiments indicated that the pike never used refuges (consistent with previous work, see Turesson and Brönmark, 2004), so they were not further provided during the actual experiment”.

2. Please discuss the limitations of 2D versus 3D tracking. Although the manuscript includes some arguments for why 2D tracking is sufficient, one of the reviewers noted that the primary supporting citation is for a prey-focused study design for different predator and prey species. Rather than making the assumption that the third dimension does not matter, please evaluate critically how violations of that assumption might affect your inferences/conclusions (or motivate further work) and ensure that this discussion is incorporated in the main text to make it apparent to readers.

We apologize for the misunderstanding as it was not our intention to state the third dimension does not matter. Rather, we decided to focus on two-dimensions as the third dimension would have made it considerably more difficult, both to keep track of the individual fish in the highly cohesive schools before and during the attacks, and to be able to consider features related to the visual field of the fish. We now better clarify the shallow water as a limitation of our study, explain why shallow water was used, and discuss our expectations of what could happen in deeper water (lines 480-494): “In terms of water depth, fish were tested in relatively very shallow water. This was primarily done to be able to keep track of individual identities and compute features related to the visual field of the fish. Shiners naturally school in very shallow water conditions as well as near the surface in deeper water in the wild (Hall et al., 1979; Krause et al., 2000b; Stone et al., 2016) and also pike primarily occur in the shallow littoral zone, sometimes only a few of tens of cm deep (Pierce et al., 2013; Skov et al., 2018). Furthermore, pilot experiment showed the pike did exhibit normal swimming and attack behaviour with attack speeds and acceleration comparable to previous work (Domenici and Blake, 1997; Walker et al., 2005). Recent other work on predator-prey dynamics did not find a considerable impact of adding the third dimension to their analyses (Romenskyy et al., 2020). Still, the water depth used is a limiting factor of our study and in the future this type of work should be extended to deeper water while still keeping track of individual identities over time. We expect that adding the third dimension would not change the stealthy attack behaviour of the pike and therefore still put more central individuals most at risk, but possibly attack success would be reduced because of increased predator visibility and prey escape potential in the vertical plane, which remains to be tested.”

3. Please clarify how the present results contribute to our understanding of the evolution of collective behavior in the wild, taking into account the potentially counterintuitive observation that fish in smaller clusters experienced lower predation. And please address Reviewer #2's related question about whether repeated exposure leads to reduced schooling behavior overall.

We now included a dedicated paragraph in our discussion to better explain how our results may contribute to our understanding of the evolution of animal grouping, including of the potential evolutionary consequences of central individuals experiencing higher predation risk (lines 456-467): “Predation is seen as one of the main factors to shape the collective properties of animal groups (Herbert-Read et al., 2017) and has so far generally been seen as to drive the formation of larger, more cohesive groups that exhibit collective, coordinated motion (see e.g. Beauchamp, 2004; Ioannou et al., 2012; B. H. Seghers, 1974). Our finding that central individuals are more at risk of being predated could actually have the opposite effect, with schooling having a selective disadvantage, which over time could result in weaker collective behaviour and less cohesive schools. However, we do not deem this likely as selection is likely to be group-size dependent, as discussed above. Furthermore, our multi-model inference approach revealed that, despite more central individuals experiencing higher predation risk, being close to others inside the school was still associated with a lower risk of being targeted. As most prey experience many types of predators, including sit-and-wait predators and active predators that hunt for prey, the extent and direction of such selection effects will depend on the broader predation landscape in which prey find themselves.”

We understand that the result that predation risk was higher in the main cluster could be interpreted as suggesting that schooling may have a selective disadvantage. However, we would like to point out that the per-capita-risk would actually be lower in the schools due to the dilution of risk. As in our experiments the shiners were highly cohesive and almost always in one single large school – rather than exhibiting fission-fusion dynamics (merging and splitting of schools) – we feel our dataset is not appropriate for proper interpretations of group-size effects on predation risk. We now also clarify this point in the discussion (lines 467-471): “While the finding that pike were more likely to attack the main school may also appear to indicate a selective disadvantage to school, calculating the per-capita-risk for each individual would actually reveal it is still safest to be part of the main school. Nevertheless, as the shiners in our study rarely exhibited fission-fusion dynamics we feel our dataset is not appropriate to make proper inferences about how predation risk is linked to group size.”

Repeated exposure did not impact the packing fraction of the schools and thus did not lead to reduced schooling behaviour overall, as we also stated in the results of the original manuscript (lines 183-187): “While the shiners did not show a change in their packing fraction (median nearest-neighbour distance) with repeated exposure to the pike (F1,52 = 1.81, p = 0.185), they increasingly avoided the area directly in front of the pike’s head (Appendix 2 – Figure 1A) resulting in the pike attacking from increasingly further away (target distance: F1,52 = 45.52, p < 0.001, see Appendix 2 – Figure 1B,C).”

4. Please clarify all questions related to experimental design, animal numbers, the outcome of prey attacks, and statistical analyses, and add an explanatory figure, as suggested by Reviewer #3.

We have made considerable modifications to the manuscript to accommodate the questions related the experimental design, sample sizes, repeated testing of animals, outcome of prey attacks, and statistical analyses (see detailed changes in our responses below). We thereby put extra attention on the Experimental Procedure section in the methods (lines 590-611): “We ran the experiment with two batches of pike and shiners, with the fish of each batch tested repeatedly in the arena over time. Pike were randomly selected, but only after they had at least three rest days since the previous trial, which was decided based on pilot work that showed pike were less motivated when tested with less time between trials. For each trial, shiners were also randomly selected from the holding tanks to create groups of 40 fish. Shiners were mixed to avoid potential effects of familiarity between individuals and the associated social feedback and learning effects related to group composition. However, as we used a repeated-measures design, we made sure that all shiners in the same group had the same number of exposures to the pike by keeping shiners in separate social holding tanks based on their number of pike exposures. In total we used 20 pike and started with about 1500 shiners. Pike had a mean number of 1.84 ± 0.14 attacks per trial, with 18 trials having more than 1 successful attack. However, not all pike did always attack during the trials, and 7 pike never attacked. As those trials did not provide any data on the attacks they were excluded and the pike not used in further trials. Also the shiners were excluded for subsequent trials as their future behaviour in the assay could be influenced by having had experience with a pike that did not attack. As a result, our sample size decreased with exposure and we stopped at a maximum number of 6 exposures. Like the pike, shiners were also generally tested with three rest days between trials but sometimes groups had one or a few fish that only had two rest days. This was inevitable due to the difficulty of just being able to run a few trials each day, and sometimes pike thus not attacking. Although none of the fish had any previous experience with any experiments, during the repeated exposures both shiners and pike had time to learn about the conditions they were confronted with, thus enabling us to also investigate how this impacted predators’ attack behaviour and prey response.”

While we appreciate the suggestion by Reviewer #3 to include an explanatory figure about the experimental design, we think this would result in a rather complex figure to show the repeated exposures, holding of shiners based on exposure, and fish being excluded due to lack of attacks, so hope our revisions of the text are sufficient.

Reviewer #1 (Recommendations for the authors):Overall this paper is beautifully written and presented, with appropriate analyses.It is packed full of information, and as a result, it can be a little dense and hard to follow at times. Some streamlining, signposting and highlighting of key points would be of benefit.One solution may be clearer hypotheses or at least clearer aims. This would add some further structure to the text. As a reader, going from the Introduction to the first line in the Results section being about describing pike movements, it was a little tricky to immediately follow the logic and flow.

We thank the reviewer for their positive remarks. We have considerably revised the introduction to address the different points of the reviewer. We now better clarify the aims of our study on lines 95-98: “To advance our understanding of differential predation risk in animal groups we need to systematically investigate the different absolute and relative spatial and visual features of both prey and predator while considering the realtime dynamics between live predators attacking groups of prey they can actually capture.”

And further on lines 100-102: “Specifically, to gain a detailed mechanistic understanding of when and where predators attack groups of prey and what predicts individual predation risk and survival."

We now also better discuss previous work, e.g. lines 69-73: “Much of the focus in the literature on the spatial effects underlying predation risk has looked at centre-to-edge and front-to-back effects, partly because they are the easiest to measure, and largely concentrated only on a single or a few key potential features (but see e.g. Lambert et al., 2021; Romenskyy et al., 2020). However, it may be more likely that a spectrum of different factors shapes predation risk within groups.” Line 81: “The majority of previous research has also focused exclusively on prey behaviour.” And lines 92-94: “It therefore remains unclear what factors actually affect attack success and the probability for targeted prey to potentially survive attacks.”

Finally, we now explain in more detail what types of analyses we did and in what order (lines 108-115): “By tracking both the predator and all prey individually over time, we were able to quantify each fish’s spatial position, relative spacing, orientation, and visual field, and analysed their movement kinematics in detail throughout each attack (Figure 1). We then quantified in detail how, when, and where the pike attacked. Subsequently, we used model fitting procedures to infer, both from the prey’s and predator’s perspective, the relative importance of a suite of potential features to predict shiners’ risk to be targeted. Finally, as pike were able to catch and consume their prey, we were able to investigate what factors best predicted the likelihood of attacks to be successful and thus for prey to survive a predator attack (see Table 1).”

From reading everything, it is still not clear to me if the pike actually caught and killed the shiners? It feels a little like the ethics section dances somewhat around this point. Please clarify.

This aspect of the study, the predictors of attack success and pike being able to consume their prey, is actually one of the key novelties of our work, as most previous work only quantified the risk to be attacked or could not quantify the predictors for survival. While this was noted in the original manuscript (e.g. lines 30, 107-108, 288-304, 407-422), we have now made this more explicit and have added further clarifications throughout, including on line 114 “(pike were able to catch and consume their prey …)”, line 294 “(The pike in our study were allowed to catch and consume their prey and were relatively successful…)”, lines 518-519 “(what predicts individual predation risk and survival)”, lines 582583 “(trials were not stopped after the first shiner was captured or eaten)”, and line 717 “(live predators could interact freely with and consume their prey).”

The authors allude to the captivity aspect in their Discussion but I think this needs to be greater. There is still limited space for the prey to move and react. You mention this, but you don't expand on what you think would happen if the prey had endless options?

We would like to note that in the methods of the original manuscript we stated our expectations regarding this point (lines 525-526): “We expect that if a much larger space would have been used, pike would still show the same approach and attack behaviour linked to their stealthy attack strategy.”

Nevertheless, following this, and the other reviewers’ suggestions, we now dedicate two new paragraphs in the discussion to the potential limitations of our experimental design, including about the limited space available. We now state more extensively (lines 472-480): “Laboratory studies on predator-prey dynamics like ours do, of course, have their limitations. Although the size of the arena we used (~2m^2^) is in line with behavioural studies with large schools of fish (e.g. Sosna et al., 2019; Strandburg-Peshkin et al., 2013) and experiments with live predators attacking schooling prey (Bumann et al., 1997; Magurran and Pitcher, 1987; Neill and Cullen, 1974; Romenskyy et al., 2020; Theodorakis, 1989), compared to conditions in the wild the prey and predator had limited space to move. However, as pike are ambush predators they tend to move relatively little to search for prey and rather rely on prey movement for encounters (Nilsson and Eklöv, 2008). Increasing tank size would have made effective tracking extremely difficult, or impossible, and while a much larger tank is expected to considerably increase latency to attack, we expect it to have relatively little effect on the observed findings.”

Do pike ambush easier when extensive cover is provided?

While we included a relevant statement about this in the methods section of the original manuscript (lines 530-533) – in which we explain that pilot experiments indicated that pike did not use refuges, in line with previous work – we have now expanded this point in the main text (lines 495-502): “For our experiments we used a testing arena without any internal structures such as refuges. This was a strategic decision as providing a more complex environment would have impacted the ability of the shiners to school in large groups and would have led fish to hide under cover. Although studying predator-prey dynamics in more complex environments would be interesting in its own regard, it would not have allowed us to study the questions we are interested in about the predation risk of free-schooling prey. Furthermore, pilot experiments indicated that the pike never used refuges (consistent with previous work, see Turesson and Brönmark, 2004), so they were not further provided during the actual experiment.”

We would like to note that, although pike are sit-and-wait predators that often hide in cover, in the wild it is predominantly younger pike that stay in vegetation, with cannibalism by larger pike being one of the main driving factors (see Skov and Lucas, 2018 Chapter 5). Due to their elongated body and narrow frontal body pike do not necessarily need refuges and are also able to ambush prey outside of cover, as our study shows. We now also better discuss this aspect on lines 338-343 of the discussion: “That predators such as pike can get very close to their prey has been described previously (Coble, 1973; Hoogland et al., 1956; Krause et al., 1998; Nursall, 1973; Webb and Skadsen, 1980) and could potentially be explained by their narrow frontal profile (Webb, 1982) as it makes it very hard for prey to detect movement changes, especially when attacked head-on. This may also explain the suggestion of previous work that pike do not suffer much from the confusion effect (Turesson and Brönmark, 2004).”

In the Introduction you state: "This not only has crucial implications for the selection of individual phenotypes, but also for the emergence of collective behaviour, and the dynamics and functioning of animal groups (Farine et al., 2015; Jolles et al.,2020; Ward and Webster, 2016). Hence, better understanding what factors influence individual predation risk within animal groups is of fundamental importance." I think this needs to be clearer. Why is it actually of fundamental importance? What real benefit is there to finding this out, beyond modelling something? Linked to this is a frequent reference to the same work, and/or the same authors. This started to feel a little like an echo chamber. I would suggest some diversification in the literature you refer too.

We agree that this was unclear. We have considerably revised the first paragraph of the introduction to better clarify the importance of improving our understanding predation risk in animal groups (lines 36-50): “A key challenge in the life of most animals is to avoid being eaten. Via effects such as enhanced predator detection (Lima, 1995; Magurran et al., 1985), predator confusion (Landeau and Terborgh, 1986), and risk dilution effects (Foster and Treherne, 1981; Turner and Pitcher, 1986), individuals living and moving in groups can reduce their risk of predation (Ioannou et al., 2012; Krause and Ruxton, 2002; Pitcher and Parrish, 1993; Ward and Webster, 2016). This helps explain why strong predation pressure is known to drive the formation of larger and more cohesive groups (Beauchamp, 2004; Krause and Ruxton, 2002; B. Seghers, 1974). However, the costs and benefits of grouping are not shared equally among individuals within groups, and besides differential food intake and costs of locomotion, group members themselves may experience widely varying risks of predation (Handegard et al., 2012; Krause, 1994; Krause and Ruxton, 2002). Where and whom predators attack within groups not only has major implications for the selection of individual phenotypes, and thereby the emergence of collective behaviour and the functioning of animal groups (Farine et al., 2015; Jolles et al., 2020; Ward and Webster, 2016), but also shapes the social behaviour of prey and the properties and structure of prey groups. Hence, a better understanding of the factors that influence predation risk within animal groups is of fundamental importance."

We now also included a dedicated paragraph about the potential evolutionary implications of our findings in the discussion (lines 456-471, see above) and included a number of new references about the role of predation in the evolution of animal grouping, including (Herbert-Read et al., 2017; Beauchamp, 2004; Seghers, 1974).

Based on the speed at which everything is happening, this can change in an instant. What time period was this considered over? Apologies if I missed this. "As most predators only have a specific region that they are biologically capable of attacking, we also only considered shiners found inside the pikes' strike zone, an area of roughly 8 cm wide and 15 cm long directly in front of the pike within which all targeted prey were positioned (Figure 2E)."

We considered the shiners that were found in the pike’s strike zone at the frame of attack but followed their position and behaviour from 0.5 s before until 0.1 s after strike initiation. For further information, please see the *Quantification of behaviour* and *Analyses section* of the Appendix.

Based on Figure 1, how relevant is the centroid?

The centroid (geometric centre of the group) is a helpful measure as it allows us to calculate the overall and relative motion of the prey group (see lines 155-156 and 198-200), the distance from which the pike attacked (lines 180-181), as well as the hierarchical clustering approach employed to establish the number of groups (lines 143150). We would like to highlight that in the manuscript we provide extensive discussion about the relevance of the group centre-edge positioning and that Figure 1 is a cropped image from a sample trial just moments before the attack that shows the relevance of the centroid in terms of the pike and shiners’ position.

How much do the lighting conditions (1) influence how pike hunt, and (2) determine how the shiners behave in response to a threat?

As pike are visual predators, in general pike display a diel activity pattern, with more activity during the day than at night. However, although quite some variation exists between populations, behavioural types, and age classes, with larger pike being more active throughout the day, pike tend to show their peaks in activity at dusk and dawn (see Skov and Lucas, 2018 Chapter 5). Hence, we expect that at relatively low light intensity pike would show the highest activity and at high light intensity the lowest. For shiners we know that a sharp reduction in light intensity results in visual impairment and thereby reduced ability to respond to their external environment. For our study design we aimed to not use too bright lights that may subdue the attack behaviour of the pike but we could also not use too dim light, both because shiners are a highly visual species and because our tracking software would struggle to differentiate the fish from the tank, especially during their fast movement during attacks. As the pike in our study tended to be active and show normal attack behaviour during the trials, we believe we found a reasonable balance with the light intensity provided. We added a clarification about the light intensity in the methods (lines 560-562): “Diffused light was provided by two LED panels positioned outside the curtains at the far ends of the tank, which provided light of medium intensity.”

And we dedicated some sentences about the potential effect of light intensity in the discussion (lines 502-508): “Experiments were conducted under artificial lights with reduces intensity but at a level high enough to be able to acquire accurate videos of the trials without motion blur. In the wild, pike are however generally found to be most active during dusk and dawn (Raat, 1988; Skov et al., 2018) and to consume most prey at low light intensity (Dobler, 1977). We expect that if a lower light intensity was used, the pike may profit from visual superiority and thereby would have increased predation success, further aided by a likely loosening of the prey schools due to limited light being available (Dobler, 1977).”

"The behavioural type of the shiners and pike was not determined but likely showed a natural range". This seems very vague and somewhat assumptive. I think there's plenty of evidence to suggest this could go either way? Please provide a little more detail here.

We apologize as we did not want to make any assumptions here about the behavioural types of the fish (i.e. consistent behavioural differences among individuals). Rather, we just wanted to clarify that was not part of our study. We have now removed the sentence to not raise any confusion. We do think that fishes behavioural type will likely play an important role in both the attack behaviour of the pike and position and avoidance response of the shiners, but this is something for future work.

"…have been of long-standing interest (Ioannou et al., 2012; Krause and Ruxton, 2002; Ward and Webster, 2016)". I'm not convinced you can say long-standing and then present references which, in the big scheme of things, are all relatively recent.

We apologise for this ambiguity. These works are reviews that cite the previous literature, and we now make this clear, preceding the citations with the term “reviewed in”. We also included Pitcher and Parrish, 1993 as a further reference.

"However, besides by predators using a stealthy approach as we show here, central individuals may also be more at risk by predators". Very clunky sentence that I had to read a few times to understand what you meant. It might benefit from a minor re-write.

We thank the reviewer for highlighting this. We have rewritten the sentence accordingly (lines 349-354): “Furthermore, increased risk of individuals near the centre of groups may be more widespread than currently thought. Predators not only exhibit stealthy behavioural tactics that enable them to approach and attack central individuals, as we show here, but may also do so by attacking groups from above (Brunton, 1997) or below (Clua and Grosvalet, 2001; Hobson, 1963; but see Romey et al., 2008), and by rushing into the main body of the group (Handegard et al., 2012; Hobson, 1963; Parrish et al., 1989).”

Reviewer #2 (Recommendations for the authors):1. The second paragraph in the introduction came off as more of a list of conflicting evidence rather than explaining a topic. Perhaps it could be restructured to better motivate the following paragraphs by stating something like "Previous work has identified many different, and sometimes contradictory, factors that predict prey vulnerability in a collective group, likely because predation strategy has been overlooked."

We thank the reviewer for this fair suggestion. We now start the second introductory paragraph with the sentence: “Previous work has identified many different, and sometimes contradictory, factors that predict prey vulnerability in groups.”

2. What was the top speed of the shiners? This might be relevant to report in line 145.

The maximum speed of the shiners was observed during the predatory attacks, which is reported on lines 126-127 of the original manuscript. We now report the 90% quantile of the top speed of the school outside of the actual attack as representative measure (line 155-156): 90% quantile: 16.3 cm/s.

3. In line 170, it might also be useful to refer to the distances between the pike and the centroid of the school in terms of the percent of total size.

The schools not necessarily had a circular or oval shape so a straight-forward computation of relative distance based on total school size is not possible. This is also one of the reasons why we further investigate if pike actually entered the groups, see results on lines 190-200.

4. Line 142: Consider changing “compromised” to “comprised”.

Corrected.

5. Line 337: Consider changing “However, besides by predators” to “In addition to predators”.

We have rewritten the sentence to correct this.

6. The topics of the paragraphs starting on lines 346 and 419 are overlapping. Perhaps these paragraphs can be combined so that the results of the study and the previous work in the field can be discussed together?

We thank the reviewer for bringing this up. We agree this is a great suggestion and we have now integrated and considerably rewritten the two paragraphs mentioned.

7. The sentence starting on line 434 is very long and confusing. Consider rewording.

The sentence starting on line 434 is relatively short while the one starting on line 442 is relatively long, so we presume that is the sentence the reviewer is referring to and rephrased it accordingly: “Furthermore, increased risk of individuals near the centre of groups may be more widespread than currently thought. Predators not only exhibit stealthy behavioural tactics that enable them to approach and attack central individuals, as we show here, but may also do so by attacking groups from above (Brunton, 1997) or below (Clua and Grosvalet, 2001; Hobson, 1963; but see Romey et al., 2008), and by rushing into the main body of the group (Handegard et al., 2012; Hobson, 1963; Parrish et al., 1989).”

8. The sentence starting on line 452 is also very long and unfocused. Consider rewording.

We thank the reviewer for pointing this out and have revised the sentence and split it into two, and moved it down in the discussion (lines 508-513): “While it is now increasingly possible to obtain detailed data from predation events on grouping prey in the field (see e.g. Handegard et al., 2012; Krause et al., 2017), even with the most sophisticated field-based imaging it would not have been possible to acquire the highly detailed data we obtained here. That is, individual-level characteristics of predator and all grouping prey throughout predator attacks at high spatial and temporal resolution, linked to attack success and survival.”

9. Consider breaking up the sentence starting on line 468: “…the predator. Even central individuals have a cha”ce".

Revised accordingly.

10. Line 575: What does "we assured perfect tracks" mean?

We apologize for the confusion. We have rephrased the sentence and instead now explain what we meant: “For our analyses we manually checked for constancy of individual shiners’ identities in the large schools at 120 fps from one second before strike initiation through to one second later, and manually corrected identity swaps and centroid and heading positions where needed.”

Reviewer #3 (Recommendations for the authors):Overall, I enjoyed reading this impressive paper. It was well written and provided a nice example of the modern, highly quantitative analyses of predator-prey attack dynamics in controlled laboratory conditions. I had only a few relatively minor comments.The authors did highly detailed analyses of metrics that explained which prey individual was targeted by the predator, and whether the targeted individual successfully evaded capture or not. This assumes that it is clear cut that there was a particular individual targeted and that the analysis correctly identified the 'targeted' individual. Given that there were probably some, if not many, attacks where multiple prey were directly in front of the predator and close by, what exact criteria did the authors use to identify which prey was targeted by the predator? In the 70% of attacks where a prey fish was captured, it is reasonably clear that that was the prey that was targeted, but in the other 30%, it seems less clear cut. Were there some, perhaps quite a few, situations where it was not 100% clear which prey was targeted? Conventional wisdom includes the idea that attack success is reduced if predators get 'confused' by having numerous prey that are potential targets, particularly if they escape in different directions. If the authors mis-identified which prey was targeted this could presumably affect the accuracy of their analyses on factors that affect attack success.

We thank the reviewer for this important question. We now include a clear explanation for how we determined the identity of the targeted individual in the methods (lines 622-627): The identity of the targeted individual was acquired by carefully comparing the video and visualisations of the tracking data at the moment of the attack. For the far majority of unsuccessful attacks we also acquired the identity of the targeted shiner with high certainty because the pike attacked the prey from very near and exhibited a clear attack trajectory. In the few cases there was a second potential target we were able to determine the most likely target by carefully going back and forth through the video, frame-by-frame, at the moment of the attack.

Our data suggests pike may not suffer much from the confusion effect, which was also suggested by Turesson & Brönmark (2004). We now further discuss this point in the context of pike being able to get so close to their prey and thereby able to actually focus on a single prey (lines 338-343): That predators such as pike can get very close to their prey has been described previously (Coble, 1973; Hoogland et al., 1956; Krause et al., 1998; Nursall, 1973; Webb and Skadsen, 1980) and could potentially be explained by their narrow frontal profile (Webb, 1982) as it makes it very hard for prey to detect movement changes, especially when attacked head-on. This may also explain the suggestion of previous work that pike do not suffer much from the confusion effect (Turesson and Brönmark, 2004).

Presumably to facilitate having sharp video images and 2-dimensional analyses of encounters, the trials were done in 6 cm of water which is likely much shallower than natural conditions for pike and shiners. The authors suggest that shiners are commonly in flat schools, so quantifying their position and movements in 3 dimensions is not necessary. They also cite a recent paper (Romenskyy et al. 2020) that suggested that the third dimension (variation in the vertical-depth axis) is not critically important in a similar study quantifying the details of attack, but this other study was on different predator and prey fish species, and focused only on which prey are targeted, not on attack success. Overall, the notion that it is valuable to study fish predator-prey interactions in 2 dimensions sounds plausible at a rough level, but I am somewhat skeptical of that conclusion at the very fine, detailed scale of this paper's analyses. Nonetheless, I found this study to be impressive and valuable, even if the detailed dynamics under some, perhaps many natural conditions (deeper water, with some refuge not far away, in different light conditions etc.) might be moderately different.

We appreciate the reviewer’s concerns about the two-dimensionality of our study. We would like to clarify that we do not think or state the third dimension is not necessary. Rather, we decided to focus on two dimensions only in this study to be able to acquire the highly detailed individual-level characteristics and quantify features related to the visual field of the fish. We now clarify this on lines 480-482: In terms of water depth, fish were tested in relatively very shallow water. This was primarily done to be able to keep track of individual identities and compute features related to the visual field of the fish. And lines 489-491: Still, the water depth used is a limiting factor of our study and in the future this type of work should be extended to deeper water while still keeping track of individual identities over time.

We now also better explain that both shiners and pike tend to spent a lot of time in shallow water, and consider how the water could have restricted the pike movements (lines 480-487): Shiners naturally school in very shallow water conditions as well as near the surface in deeper water in the wild (Hall et al., 1979; Krause et al., 2000b; Stone et al., 2016) and also pike primarily occur in the shallow littoral zone, sometimes only a few of tens of cm deep (Pierce et al., 2013; Skov et al., 2018). Furthermore, pilot experiment showed the pike did exhibit normal swimming and attack behaviour with attack speeds and acceleration comparable to previous work (Domenici and Blake, 1997; Walker et al., 2005). Pike had a body height between 3-4cm, which we now explain on lines 562-564 in the methods: The tank was filled with water to a depth of 6 cm, about 1.5x – 2x the body height of the pike. The vertical space was enough for shiners to swim over the (back of the) pike, which we indeed observed occasionally.

We removed the short discussion about the 2D aspect of our study from the methods and now also include a short discussion with our expectations for testing in three dimensions (lines 491-494): We expect that adding the third dimension would not change the stealthy attack behaviour of the pike and therefore still put more central individuals most at risk, but possibly attack success would be reduced because of increased predator visibility and prey escape potential in the vertical plane, which remains to be tested.

The experimental design and sample sizes with regard to pattern of re-use of animals across multiple days (lines 544 ff.) might be easier to understand if the authors included a summary figure. How many groups of 40 shiners began each batch? There were 9 pike in batch 1 and 4 in batch 2, does that mean that the experiment began with 360 shiners in batch 1 and 160 in batch 2? Trials continued until there were not enough fish left for another trial with that number of exposures, which I think means until there were fewer than 40 fish. But only 70% of 125 attacks were successful (about 88). If the study started with 520 prey and only 88 were killed, it sounds like the study could go through a very large number of repeated trials before it would not have enough prey to continue. I am clearly not understanding something about the design. Pike had 3 rest days between trials. Was the same rest schedule applied to shiners? Shiners were kept in tanks with fish with the same number of previous exposures, but did this mix animals with different numbers of days since their previous trial? How many shiners participated in the study? There were a total of 125 attacks, with 1.84 attacks/trial which seems to mean 68 trials, spread across 13 pike in 2 batches. But 'a number of pike' never attacked any prey. Does the 13 pike include these or are the 13 pike only the ones that did attack prey?

We thank the reviewer for asking these detailed questions about our sample size and about the repeated testing of the fish. This made us realize we weren’t clear enough about these points. In total we started with around 1500 shiners and 20 pike. Out of these 7 pike never attacked and a number of them stopped attacking during later trials, resulting in us to exclude that trial and not use those pike and shiners for any further trials. Hence, the number of attacks we acquired decreased quickly with exposure number. Like the pike, shiners were also generally tested with three rest days between trials but sometimes groups had a couple fish that only had two rest days. This was inevitable due to the difficulty of just being able to run a few trials each day, and sometimes pike not attacking.

We have considerably rewritten the Experimental Procedure section to accommodate all of the reviewers questions and help better explain our sample size and retesting of animals across the experiment (lines 589-610): We ran the experiment with two batches of pike and shiners, with the fish of each batch tested repeatedly in the arena over time. Pike were randomly selected, but only after they had at least three rest days since the previous trial, which was decided based on pilot work that showed pike were less motivated when tested with less time between trials. For each trial, shiners were also randomly selected from the holding tanks to create groups of 40 fish. This group size reflects the size of shiner shoals observed in the wild (20; 41). Shiners were mixed to avoid potential effects of familiarity between individuals and the associated social feedback and learning effects related to group composition. However, as we used a repeated-measures design, we made sure that all shiners in the same group had the same number of exposures to the pike by keeping shiners in separate social holding tanks based on their number of pike exposures. In total we used 20 pike and started with about 1500 shiners. Pike had a mean number of 1.84 ± 0.14 attacks per trial, with 18 trials having more than 1 successful attack. However, not all pike did always attack during the trials, and 7 pike never attacked. As those trials did not provide any data on the attacks they were excluded and the pike not used in further trials. Also the shiners were excluded for subsequent trials as their future behaviour in the assay could be influenced by having had experience with a pike that did not attack. As a result our sample size decreased with exposure and we stopped at a maximum number of 6 exposures. Like the pike,shiners were also generally tested with three rest days between trials but sometimes groups had a couple fish that only had two rest days. This was inevitable due to the difficulty of just being able to run a few trials each day, and sometimes pike thus not attacking. Although none of the fish had any previous experience with any experiments, during the repeated exposures both shiners and pike had time to learn and anticipate about the situation, thus enabling us to also investigate how this impacted predators’ attack behaviour and prey response.

I will admit that I might not have outlined the paper carefully enough, but the statistical analyses described briefly in lines 193 ff., and in some detail in the Appendix involved reasonably standard model choice methods listing relative model weights, but in the main text, the authors also showed various other statistics: LRT chi-square values, F-values, with pvalues, R-squares, and correlation coefficients. Perhaps I missed where they described these, but if indeed they did not do so, they should describe all statistical tests.

We carefully checked all results and description of the analyses again and now state more clearly in the methods how all statistics were acquired in the few cases where this was still missing, on lines 703-706: To investigate how repeated exposure changed the shiners packing fraction and avoidance of the pike’s head and pikes’ distance from the targeted individual, we ran a linear model with exposure, fitted as a cubic function, as a fixed factor and data subsetted to pikes’ first attack attempt during the trials. And lines 709- 711: Thereby estimates, likelihood-ratio tests, and p-valuesreported in the text were acquired from the final models. In some cases we included Pearson correlations and linear models to further investigate the relationship and explained variance (R2 ) between two predictor variables.

Model choice analyses also allow one to assign relative weights not just to each model, but to each factor by summing the weights for each model in which that variable appears. Rather than only show the top 15 models (Appendix 3), it might be useful to calculate the weights for each factor. This would complement the results and conclusions in the main text.

For our analyses into what variables best predicted individual predation risk and survival we did just that: we selected the final predictive features based on both their relative importance score across all models, as the reviewer is suggesting, as well as by considering the features that appeared in the most parsimonious models. Plots of the relative weights of each factor are given in the main text (panel A in figures 4-6).

Perhaps I did not study the Appendices carefully enough, but the best listed models (Appendix 3 and 4) did not include number of exposures, pike ID or attack attempt, or more importantly, interactions between these random effects and the predictive features from either the prey or predator view. Appendix 2 shows that exposure number had large effects on pike distance from the group centroid and from the target individual. Did the more detailed features of targeted prey (or attack success) change with repeated exposures? For each exposure number (Appendix 2 - figure) the distances varied quite a lot. I assume that these are for the 13 different pike. Did individual pike differ consistently in their targeting tendencies, or even in their change in targeting tendencies with repeated exposure and learning? If so, this deserves more discussion.

In our models we always included exposure number, pike ID and attack attempt as random effects, as we stated in the original manuscript on lines 963-965. They are not features that help us better predict which individual may be targeted or survive an attack but factors that may influence the variation among trials. Because of that they are not included in the weighting of the models nor as any interaction with the fixed factors. We did not run further detailed analyses with repeated exposure to see how the importance of different predictive features may change with repeated exposure as this is not the focus of the present study and the amount of results we present is already quite dense, as Reviewer 1 also pointed out. We saw that when exposure was included as a random factor it played only a very minor role in explaining the variance among trials. Although the question if pike differed in their change in targeting with repeated exposure and learning is definitely interesting, this again goes beyond the focus of the presentstudy. A proper investigation of such questions about how predators may learn and adapt their attack tendencies with experience warrants a carefully designed study with that focus in mind.

Appendix 2, figure 1 -- what are the darker points in figures B and C?

Semi-transparent points indicate individual trials. Some points appear darker due to overlap. We now clarify this in the figure legend.

Line 193: 'were' not 'where'.